# Progress in the Application of Nanomaterials in Tumor Treatment

**DOI:** 10.3390/biomedicines13112666

**Published:** 2025-10-30

**Authors:** Xingyu He, Lilin Wang, Tongtong Zhang, Tianqi Lu

**Affiliations:** 1Obesity and Metabolism Medicine-Engineering Integration Laboratory, Department of General Surgery, Affiliated Hospital of Southwest Jiaotong University, The Third People’s Hospital of Chengdu, Chengdu 610000, China; 2School of Life Science and Engineering, Southwest Jiaotong University, Chengdu 610000, China; 3Medical Research Center, Affiliated Hospital of Southwest Jiaotong University, The Third People’s Hospital of Chengdu, Chengdu 610000, China

**Keywords:** nanomaterials, cancer therapy, combination therapy

## Abstract

Cancer continues to pose a major global health burden, with conventional therapeutic modalities such as surgical resection, chemotherapy, radiotherapy, and immunotherapy often hindered by limited tumor specificity, substantial systemic toxicity, and the emergence of multidrug resistance. The rapid advancement of nanotechnology has introduced functionalized nanomaterials as innovative tools in the realm of precision oncology. These nanoplatforms possess desirable physicochemical properties, including tunable particle size, favorable biocompatibility, and programmable surface chemistry, which collectively enable enhanced tumor targeting and reduced off-target effects. This review systematically examines recent developments in the application of nanomaterials for cancer therapy, with a focus on several representative nanocarrier systems. These include lipid-based formulations, synthetic polymeric nanoparticles, inorganic nanostructures composed of metallic or non-metallic elements, and carbon-based nanomaterials. In addition, the article outlines key strategies for functionalization, such as ligand-mediated targeting, stimulus-responsive drug release mechanisms, and biomimetic surface engineering to improve in vivo stability and immune evasion. These multifunctional nanocarriers have demonstrated significant potential across a range of therapeutic applications, including targeted drug delivery, photothermal therapy, photodynamic therapy, and cancer immunotherapy. When integrated into combinatorial treatment regimens, they have exhibited synergistic therapeutic effects, contributing to improved efficacy by overcoming tumor heterogeneity and resistance mechanisms. A growing body of preclinical evidence supports their ability to suppress tumor progression, minimize systemic toxicity, and enhance antitumor immune responses. This review further explores the design principles of multifunctional nanoplatforms and their comprehensive application in combination therapies, highlighting their preclinical efficacy. In addition, it critically examines major challenges impeding the clinical translation of nanomedicine. By identifying these obstacles, the review provides a valuable roadmap to guide future research and development. Overall, this work serves as an important reference for researchers, clinicians, and regulatory bodies aiming to advance the safe, effective, and personalized application of nanotechnology in cancer treatment.

## 1. Introduction

In recent years, cancer (tumor diseases) has become a common disease and one of the main causes of death in the world [1]. Demographic-based forecasts indicate that by 2050, the number of new cases of malignant tumors is expected to exceed 35 million per year [2]. Improving the compliance of tumor therapy and increasing the cure rate of malignant tumors are of great significance in enhancing the overall quality of life and health of the population. At present, the core strategies of tumor treatment still focus on chemotherapy, radiotherapy, surgical resection, and immunotherapy, which have emerged in recent years [3]. However, the surgical complications, adverse reactions of radiotherapy and chemotherapy, and drug resistance of tumor cells associated with this series of therapies have greatly weakened the quality of life of tumor patients, seriously restricting their widespread application and efficacy improvement. With the development of nanomaterials and the diversification of their functions, their application in cancer treatment has gradually become more extensive. Nanomaterials can not only be used as an independent therapeutic drug, but also as a delivery carrier for various anti-cancer drugs. They can deliver drugs to tumor sites in a targeted manner and increase the enrichment concentration of drugs at tumor sites, thereby achieving multiple functions such as targeted delivery, enriched drug delivery, and synergistic anti-cancer effects. It is expected to provide a new idea for clinical cancer treatment.

Currently, clinical cancer treatment relies on surgical resection, chemotherapy [4,5,6], radiation therapy [7], and the rapidly advancing immunotherapy [8,9]. However, these conventional therapies all exhibit significant limitations: Surgery may carry risks of complications and recurrence; Chemotherapy is associated with severe adverse reactions such as hair loss, bone marrow suppression, and drug resistance [10]; Radiotherapy may cause damage to normal tissues and inadequate control of distant metastases [11]. Immunotherapy shows promising prospects but involves individual variability and immune-related toxicity [12,13].

Chemotherapy has become the main method for clinical treatment of cancer by using cytotoxic drugs to inhibit or kill rapidly dividing tumor cells. Typical drugs include doxorubicin [6], paclitaxel [14,15], and daunorubicin [16], among others, which mainly interfere with the cell division cycle or DNA synthesis process [17]. Although there are many types of tumor treatment drugs, their common feature is that the drug treatment effect is highly dependent on the drug dose that enters the tumor cells to exert its impact [4,5]. Chemotherapy cannot accurately identify cancer cells, thereby affecting normal, actively dividing cells, leading to toxic side effects such as hair loss, bone marrow suppression, and gastrointestinal damage [18,19]. Long-term use of drugs can induce multidrug resistance mechanisms in cancer cells, reducing the efficacy [10].

Radiation therapy, also known as radiotherapy (RT), is currently one of the most commonly used cancer treatment methods. RT causes DNA damage in tumor cells through high-energy rays, thereby inducing apoptosis [7]. Inflammatory factors released by irradiated cells can regulate immune cells in the tumor microenvironment (TME) and further kill tumors. In addition, tumor-associated antigens produced by irradiated tumor cells can be presented to T cells by DCs in the TME, activating systemic anti-tumor immune responses. However, RT is a double-edged sword, and its clinical effects are contradictory. Irradiation kills tumor cells, but it also kills immune effector cells and damages normal tissues. When the radiation dose exceeds the repair capacity of surrounding normal tissues, it will cause irreversible damage. Moreover, RT is only effective in the irradiated area, making it difficult to control distant metastases. Normal tissues within the irradiated range will also be damaged, which may lead to sequelae such as radiation pneumonia and fibrosis [11].

Surgical resection is one of the oldest and most direct treatment methods. It aims to completely remove the tumor and the surrounding tissues that may be invaded by surgical means; that is, there are no residual cancer cells under the microscope. During the operation, the surgeon will choose the appropriate range and method of resection according to the type, size, and location of the tumor and the overall health of the patient. It is especially suitable for solid tumors without metastasis in the early stage. However, even if the primary lesion is completely removed, recurrence may still occur due to hidden micrometastatic cells [20]; surgical resection is difficult to accurately define the tumor margin, which may cause residual tumors. At the same time, surgery may cause complications such as infection, bleeding, and damage to adjacent organs. Studies have shown that postoperative infectious complications can significantly increase the risk of recurrence and mortality in patients [21,22]; certain surgeries, such as lobectomy, may cause chronic postoperative pain syndrome, manifested as persistent chest pain, dyspnea, and other symptoms, seriously affecting the patient’s quality of life [23].

The goal of tumor immunotherapy is to stimulate the host’s immune system or directly deliver tumor-targeted killer immune cells to enable them to produce immune attacks on malignant tumors and thus play a role in fighting tumors [8], therapeutic strategies such as oncolytic virotherapy, cancer immunization, cytokine-based treatments, adoptive transfer of immune cells, and blockade of immune checkpoints [9]. The limitation is that there are obvious individual differences in immunotherapy, and the response rates of different tumor types or different patients vary greatly [12], which can lead to autoimmune complications such as enteritis, pneumonia, and endocrine disorders, all of which reflect systemic immune dysregulation and may significantly compromise patient outcomes [13]. CAR-T therapy is extremely expensive and difficult to widely popularize.

Traditional tumor treatment methods provide important support for clinical treatment, but their inherent limitations indicate that future tumor treatment should be more precise, efficient, and less toxic. In recent years, nanomaterials have received extensive attention in tumor treatment due to their high drug loading potential, biocompatibility, and drug stability [24], and are expected to break through the current difficulties and improve the individualization and systemic level of tumor treatment [25]. Nanomedicine refers to the application of nanotechnology in medicine. Its research in tumor treatment mainly focuses on the design, manufacture, regulation, and application of nanomedicine composed of particles < 100 nm [26]. The small size and large area-to-volume ratio of nanoparticles enable them to efficiently bind, absorb, and deliver small-molecule drugs, DNA, RNA, proteins, and probes. Their variable size, shape, and surface characteristics also give them high stability, high carrying capacity, the ability to bind hydrophilic or hydrophobic substances, and compatibility with different routes of administration [27]; they facilitate interactions with biological molecules or cells and enhance biological activity. Numerous nanomaterials, such as liposomes, polymeric nanoparticles, and other biocompatible systems, have exhibited favorable biocompatibility and reduced toxicity to normal tissues [28]. In recent years, nanomedicine has made significant progress and development in many fields [29]. Nanomaterials can be used as carriers of chemotherapy drugs to improve the therapeutic index of drugs and can also be used as drug carriers for immunotherapy, microbial therapy, gene therapy, photothermal, and optogenetic therapy in clinical tumor treatment. The versatility of nanomedicine products can integrate drug delivery, diagnostic imaging, and treatment, and also provide more possibilities for treatment [30]. Currently, some nanomedicine platforms have been developed for clinical cancer treatment. As shown in the figure (Figure 1), new cancer treatment methods have been developed using various nanomaterials such as lipid-based and polymer-based nanocarriers, inorganic nanoparticles, viral nanoparticles, and drug conjugates [31].

As nanotechnology continues to advance within biomedicine, its application in cancer diagnosis and treatment is increasingly overcoming the inherent limitations of conventional therapeutic approaches. Nanotechnology has developed extensively and rapidly in the medical field. The application of nanomaterials in tumor diagnosis and treatment has broken the inherent limitations of traditional cancer diagnosis and treatment methods. Nanomedicine combines nanoscience and technology with pharmaceutical characteristics to achieve multifunctional and multimodal treatment, which can improve drug efficacy and traceability. In comparison to conventional anticancer agents, nanomedicines exhibit distinct advantages in several key aspects [32]: (1) Nanomedicines can improve solubility and enhance chemical stability [33,34]. (2) Nanomedicine can improve pharmacokinetics and prevent biodegradation or excretion of anticancer drugs [35]. (3) Nanomedicine can help improve the distribution and targeting of anticancer drugs, and this targeted treatment is likely to reduce tumor resistance to anticancer drugs [36,37]. (4) Nanomedicine can also be designed to release drug carriers when triggered, forming a stimuli-sensitive treatment method, which can greatly increase intracellular drug uptake and intracellular drug release [38].

Despite the tremendous potential of nanomaterials in cancer therapy, their clinical translation still faces a range of critical challenges. These include the incomplete evaluation of their biosafety and long-term toxicity [39]; limited understanding of their in vivo biodistribution, metabolism, and clearance mechanisms [34]; and the lack of scalable, controllable, and reproducible manufacturing technologies for clinical-grade nanomaterials [40]. Moreover, the intrinsic heterogeneity of the tumor microenvironment, along with interindividual biological variability among patients, further complicates the broad applicability and therapeutic consistency of nanomedicine-based strategies [32].

In response to these challenges, this review aims to systematically summarize recent advances in the application of nanomaterials for cancer treatment. To achieve this objective, the search and research strategy adopted in this paper is as follows:

Database Search: Conduct systematic searches in PubMed, Web of Science, Scopus, and CNKI databases using core keywords such as “nanoparticles”, “tumor/cancer therapy”, “nanocarriers”, and “clinical translation”. Refine selections by combining carrier-specific keywords like “liposome”, “polymeric nanoparticles”, and “inorganic nanomaterials”.

Search Timeframe: Limited to English and Chinese literature published between January 2013 and May 2024 to ensure coverage of the latest research advances over the past decade.

Inclusion and exclusion criteria: Original research papers, reviews, and clinical studies focusing on the application and clinical translation of nanomaterials in tumor chemotherapy, photothermal therapy, immunotherapy, and related fields were included. Duplicate publications, conference abstracts, and studies with significant experimental design flaws were excluded.

We highlight the design principles and functionalization strategies of various nanocarriers, including liposomes, polymeric nanoparticles, inorganic nanomaterials, and carbon-based nanostructures. Furthermore, we explore their applications across major therapeutic modalities, such as chemotherapy, photothermal therapy (PTT), photodynamic therapy (PDT), gene therapy, and immunotherapy. In addition, this review provides an in-depth analysis of the key translational barriers to clinical application, including concerns over biocompatibility, pharmacokinetic behaviors, difficulties in industrial-scale production, and regulatory compliance. Building upon this foundation, we propose future research directions that emphasize the importance of interdisciplinary innovation, particularly the integration of nanotechnology, intelligent nanoplatforms, and sustainable green approaches. By addressing current bottlenecks and leveraging technological advancements, nanomaterials are expected to drive the development of more efficient, safer, and personalized cancer therapeutic strategies. This review intends to serve as a comprehensive reference and forward-looking perspective for researchers and clinicians engaged in the field, intending to facilitate the clinical translation and broader application of nanomedicine in oncology.

This review uniquely focuses on research advances in nanomaterials for tumor therapy from 2013 to 2025, covering fundamental research, preclinical validation, and clinical translation exploration within a clearly defined scope. It not only evaluates nanomaterials’ application as carriers for chemotherapy and immunotherapy but also discusses their potential in regulating the tumor microenvironment and multimodal therapy. While summarizing research outcomes, it particularly highlights challenges in nanomaterial safety, scalable production, and regulatory compliance, while maintaining a rational clinical translation orientation. It proposes multidisciplinary integration through smart nanoplatforms, green sustainable synthesis, and AI-assisted design to advance nanomedicine toward clinical translation and widespread application.

## 2. Common Types of Nanomaterials and Their Characteristics

### 2.1. Organic Nanomaterials

#### 2.1.1. Liposomes

Among various nanomaterials, liposomes are the earliest discovered nanocarriers and remain the most extensively studied and widely applied in targeted drug delivery systems. First identified in the 1960s by Bangham et al., liposomes have since become one of the most commonly utilized nanocarriers for targeted drug delivery [41]. Liposomes are spherical lipid bilayers with a diameter of 50 to 1000 nm. They can be used as promising drug delivery carriers for bioactive compounds (Figure 2), mainly by reducing the toxic effects of drugs when used alone [42]. Due to their amphiphilic nature, lipophilic drugs can be encapsulated within the phospholipid bilayer or adsorbed onto the liposomal surface, while hydrophilic drugs can be enclosed within the aqueous core of the vesicle. Moreover, liposomes exhibit excellent biocompatibility and biodegradability owing to their phospholipid bilayer structure. Liposomes have been widely employed as carriers for small-molecule drugs, peptides, proteins [43], and nucleic acids [44]; With advantages such as low toxicity and immunogenicity, excellent biocompatibility, and an easily modifiable surface, liposomes, owing to their cell membrane–like structure—demonstrate favorable biological compatibility and have been widely applied in the delivery of various anticancer drugs [45]. In recent years, liposomal drug delivery systems have made significant progress, markedly improving drug sustained-release profiles and therapeutic indices. To date, several liposome-based nanopharmaceuticals have been approved by the U.S. Food and Drug Administration (FDA), including liposomal daunorubicin, doxorubicin, and irinotecan [46,47].

Liposomes possess tunable characteristics such as particle size, surface charge, and membrane permeability, which collectively influence their biodistribution, cellular uptake efficiency, and drug release behavior. Particularly in cancer therapy, liposomes can exploit the enhanced permeability and retention (EPR) effect—characterized by leaky tumor vasculature and impaired lymphatic drainage—for passive targeting and accumulation at tumor sites [48]. Additionally, surface modification with polyethylene glycol (PEG) to form “long-circulating liposomes” can significantly prolong their plasma half-life and reduce clearance by the reticuloendothelial system, thereby enhancing therapeutic efficacy [49].

To overcome the limited specificity of conventional liposomes, various active targeting strategies have been developed. Common approaches include covalent attachment of targeting ligands, such as folic acid, antibodies, peptides, or carbohydrates, on the liposomal surface to recognize overexpressed receptors on tumor cells, including HER2, EGFR, and CD44 [50]. In a demonstration of this principle, investigations have revealed that liposomes functionalized with anti-HER2 antibodies exhibit a marked enhancement in their affinity for breast cancer cells, concomitantly augmenting the cytotoxic efficacy of doxorubicin [51]. In recent years, stimuli-responsive liposomes have garnered increasing attention. These systems are designed to release their payloads in response to specific triggers in the tumor microenvironment, such as acidic pH, redox conditions, or enzyme activity. For instance, pH-sensitive liposomes incorporate pH-labile lipids or chemical linkages to enable rapid drug release in the acidic conditions of tumor tissues or lysosomes, thereby improving targeting specificity and reducing systemic toxicity [52].

Liposomal surfaces can be modified through a variety of strategies to endow multifunctionality, including prolonged systemic circulation, enhanced accumulation in target tissues, improved cellular internalization, and organelle-specific drug delivery [53]. New liposomes, such as targeted modified liposomes, stimulus-responsive liposomes, and biomimetic liposomes, are emerging one after another and are a hot topic in the research of new anti-tumor drug preparations [54]. At present, some preparations with liposomes as carriers have been successfully used in clinical practice, such as doxorubicin hydrochloride liposomes and amphotericin B liposomes [55].

#### 2.1.2. High Polymer Nanoparticles

Cationic polymers are capable of condensing genetic material into nanoparticles, typically ranging from tens to hundreds of nanometers in diameter. This is achieved through electrostatic or hydrophobic interactions between their abundant positively charged groups, DNA, or RNA [56]. Moreover, the cationic surface groups facilitate binding to negatively charged cell membrane components, thereby promoting cellular internalization and offering enhanced resistance to enzymatic degradation [57]. The structures of polymer nanoparticles are diverse, mainly including nanoparticles, nanofibers, nanofilms, and nanocapsules. Polymeric nanostructures exhibit diverse architectures, including nanoparticles, nanofibers, nanomembranes, nanocapsules, and polymeric micelles [58]. For example, researchers have noted that nanoparticle polymer micelle paclitaxel represents a novel paclitaxel nanoparticle micelle formulation free of Cremophor EL. Composed of amphiphilic block copolymers featuring a hydrophilic shell and hydrophobic core, it encapsulates paclitaxel within its structure. Potential advantages of this polymeric micelle include enhanced paclitaxel solubility, avoidance of Cremophor EL-related toxicities such as severe allergic reactions and neuropathy, while potentially offering improved antitumor efficacy and safety [59]. Other studies describe CRLX101 as a dynamic tumor-targeted nanomedicine containing cyclodextrin polymer (CDP) conjugated to camptothecin (CPT). It selectively targets tumor cells, prolongs the release of active CPT, and enhances antitumor activity. This formulation has entered and completed Phase I/II clinical evaluation with a clear clinical development pathway. early Phase II clinical evaluation with a defined clinical development pathway [60]. They usually have a porous structure inside, which can effectively encapsulate and protect bioactive substances and ensure stability and biocompatibility when released in vivo [61]. Polylactic-co-glycolic acid (PLGA) is one of the most representative synthetic polymers, mainly composed of glycolic acid and lactic acid monomers. The polymer exhibits complete biodegradability in aqueous media, and the degradation performance can be optimized by adjusting the chemical composition and polymer chain length [62]. In addition, specific sites in PLGA nanocarriers can significantly enhance the solubility of hydrophobic anticancer peptides, thereby improving their bioavailability [63]. Girgis et al. [64] synthesized PLGA nanoparticles by microfluidics technology and successfully delivered the novel Ran inhibitory peptide CK-10 to MDA-MB-231 breast cancer cells, effectively inhibiting the proliferation and metastasis of cancer cells. To overcome the instability and toxicity of cationic ACPs in vivo, Yang et al. [65] used PEG to condense cationic ACPs into nanoparticles to neutralize the positive charge of ACPs. The results showed that the complex effectively killed tumor cells by inducing tumor cell apoptosis and was able to effectively treat doxorubicin-resistant MCF-7 cells (Figure 3).

#### 2.1.3. Peptide Nanoparticles

Peptides are a class of compounds composed of three or more amino acid molecules that form peptide bonds through dehydration condensation. Peptides combined with drugs can be used to develop drug delivery systems to improve the intensity and accuracy of drug treatment. They can also be used as carriers to deliver anti-tumor drugs to specific sites and improve the delivery ability of DDS [66]. Notably, peptide-based carriers significantly improve the intracellular delivery efficiency and target specificity of therapeutic agents, thereby effectively overcoming the challenges associated with drug transport barriers [67].

The anti-tumor mechanism of peptide nanoparticles mainly includes inducing tumor cell apoptosis and necrosis, inhibiting tumor angiogenesis, and activating anti-tumor immune response [68]. Compared to conventional nanomaterials, peptide-based nanoparticles offer distinct advantages, including structural diversity, excellent biocompatibility, controllable biodegradability, and ease of synthesis and modification. These properties make them highly applicable in the field of disease treatment, with widespread use in antidiabetic, antitumor, and antimicrobial peptide-based therapeutics [69]. Furthermore, peptides possess favorable cell-penetrating capabilities and target recognition properties. Through rational design, they can be engineered to bind specific receptors with high precision, thereby enhancing drug accumulation and therapeutic efficacy at diseased sites [70].

In cancer therapy, peptide nanoparticles enable active targeted delivery by recognizing tumor-associated receptors such as integrin αvβ3, EGFR, and HER2, which significantly improves the therapeutic index of chemotherapeutic agents while reducing systemic toxicity [71]. In addition, peptides can function as integral components of drug delivery platforms, enabling stimulus-responsive release through programmable sequence design. For instance, researchers have developed peptide-based nanoparticles incorporating pH-sensitive motifs that undergo conformational changes in the acidic tumor microenvironment, triggering drug release and facilitating precision therapy [72]. Moreover, nanoparticle carriers modified with cell-penetrating peptides (CPPs) exhibit enhanced membrane permeability, leading to increased intracellular accumulation of therapeutic agents in tumor cells. This strategy has been shown to overcome drug efflux and resistance mechanisms, thereby improving treatment outcomes [73].

It is noteworthy that peptide-grafted polymer nanoparticles and pure peptide-based nanoparticles each possess distinct advantages in performance. The former, by functionalizing peptides onto the surface of a controllably degradable polymer carrier, not only maintains the stability and extended circulation time of the polymeric nanoparticle but also enables precise delivery of targeted ligands, reducing the risk of rapid clearance in vivo [74]. However, their relatively complex structure results in higher synthesis and large-scale production costs [75]. In contrast, peptide-based nanoparticles typically form through peptide self-assembly, exhibiting high biocompatibility and programmability. They readily respond to the tumor microenvironment and enable efficient drug release [76]. However, their limited stability and susceptibility to rapid body degradation and clearance constrain clinical application. A rational combination and optimization of both approaches may represent a significant future direction in precision cancer therapy.

Thanks to the rapid development of modern molecular biology, peptide materials, as a new type of biomedical material, have shown great application potential in the field of tumor treatment. Therefore, it is of great significance to develop peptides with different functions and explore their biological activities [77,78]. It has been found that many peptide drugs have entered clinical trials or have been approved for marketing. These peptides can be used in different ways for cancer treatment as peptide hormones [79], peptide vaccines [80], cytotoxic drug carriers [81], and anticancer drugs [82].

### 2.2. Inorganic Nanomaterials

#### 2.2.1. Gold Nanoparticles

PTT uses the thermal energy induced by the laser of a photothermal agent to achieve thermal ablation of tumor cells. It can kill tumor cells at high temperatures. The efficacy of PTT mainly depends on the selection of photothermal conversion agents. In recent years, gold nanoparticles have emerged as one of the most extensively investigated metal nanomaterials, owing to their remarkable plasmonic optical characteristics, strong bioconjugation capabilities, excellent biocompatibility, high dispersion stability, and low cytotoxicity. Compared with other metal nanoparticles, gold nanoparticles have more advantages [83] and are currently one of the most concerned photothermal conversion agents [84].

AuNPs have a strong localized surface plasmon resonance effect [85]. This resonance phenomenon enables gold nanoparticles to absorb a large amount of light energy and efficiently convert light energy into heat energy through the non-radiative electron relaxation dynamics process. This conversion process makes metal nanoparticles a very effective heat source, which can generate local high temperatures at the tumor site, thereby killing cancer cells [86]. When gold nanoparticles are irradiated by photons, they interact with each other, causing the electrons inside the gold nanoparticles to be excited. Upon excitation, the free electrons on the surface of gold nanoparticles undergo collective oscillation and generate charge separation. This collective electron movement facilitates the efficient absorption of light energy, which is subsequently converted into heat via electron relaxation processes. Electron-electron relaxation refers to the transfer of excited-state energy to other electrons through collisions and interactions between electrons, allowing them to reach a higher energy state. Electron–phonon relaxation refers to the energy transfer from excited electrons to the metal lattice in the form of phonon excitation (i.e., lattice vibrations), which enhances vibrational motion and results in heat generation. During this process, gold nanoparticles form a thermally excited metal lattice, thereby achieving the conversion of absorbed light energy into thermal energy [87].

AuNPs are easy to synthesize and surface functionalize, allowing the coupling of therapeutic agents, targeting agents, and stabilizers on their surfaces, thereby prolonging their existence in the blood circulation and increasing their aggregation in the tumor area [88]. Raeesi et al. [89] found that photothermal therapy can destroy the extracellular matrix, clearing the way for gold nanorods to enter tumor cells. At the same time, heat generation induced by gold nanorods can further denature structures such as collagen fibers, thereby promoting the transport of more nanoparticles to diseased tissues.

#### 2.2.2. Silica Nanoparticles

Silica nanoparticles typically exist in porous forms, particularly mesoporous silica nanoparticles (MSNs), which feature a regular and tunable pore structure (generally within the range of 1.5–10 nm). This structure endows them with high specific surface area, large pore volume, and excellent loading capacity [90,91]. Precisely due to this characteristic porous nature, the advantages and limitations of silica nanoparticles in drug delivery and biomedical applications are more readily understood: their strengths lie in high drug loading capacity, controllable structure, and readily functionalizable surfaces. However, their long-term metabolism and potential accumulation effects within the body require further evaluation.

Silica is one of the most complex and abundant materials on Earth. It exists in natural minerals and can also be obtained through artificial synthesis [92]. Silica nanoparticles are widely used in various fields due to their advantages, such as simple preparation, low cost, hydrophilicity, good biocompatibility, large specific surface area, large pore volume, and controllable particle size [93]. Vallet-Regi and his collaborators first reported in the early 21st century that silicon-based nanostructured mesoporous silica nanoparticles (MSN) can also be used as potential nanocarriers for tumor-specific drug/gene delivery [90,91,94]. Researchers have successfully utilized MSNs to load small-molecule chemical compounds as well as large-molecule biological substances, including deoxyribonucleic acid (DNA), ribonucleic acid (RNA), and proteins, so as to achieve the function of targeted drug delivery [95].

In recent years, nanosilica has also been widely used in drug delivery, gene delivery, tumor imaging, bioimaging, photodynamic therapy, photothermal therapy, and immunotherapy due to its excellent structural properties and biocompatibility [96,97,98] (Figure 4). Tivnan et al. [99] reported that they developed functionalized GD2 antibody-coupled porous silica nanoparticles to deliver miR-34a to neuroblastoma tumors.

#### 2.2.3. Magnetic Nanoparticles

Owing to their distinctive physicochemical characteristics, magnetic nanoparticles have found extensive applications in both biomedical and clinical fields, such as magnetically guided drug delivery, magnetic hyperthermia for tumor treatment, and cancer immunotherapy [100,101,102,103].

At present, there are two main types of materials used to modify magnetic nanomaterials: natural biomacromolecules and synthetic polymers. Surface modification of nanoparticles can change the particle size and morphology, and accordingly improve the particle cytotoxicity and biocompatibility. The hydroxyl groups carried by the modified materials can specifically bind to antibodies, lectins, folic acid, protein peptides, hormones, nucleotides, and biotin, among others, and achieve targeted drug transport under the action of an external magnetic field, reducing the adverse reactions of drugs [104,105,106,107,108,109].

With the application of nanotechnology in biomedicine, researchers have constructed magnetic nanoparticle drug delivery systems to couple chemotherapy drugs to the surface or interior of surface-modified nanoparticles [110,111]. By applying an alternating magnetic field, the enhanced drug permeability and EPR allow more drugs to accumulate in tumors [112]. For instance, Cheng et al. [113] reported the use of gelatin to encapsulate Fe_3_O_4_ nanoparticles, carrying low-toxic platinum IV (Pt IV) prodrugs, and further improving the anti-cancer effect by releasing platinum II (Pt II) in the cell environment and integrating pancreatic enzymes to degrade gelatin.

Modern hyperthermia mostly uses microwaves and ultrasound to perform local hyperthermia on tumor lesions, but due to insufficient targeting, while destroying tumor tissue, it also causes damage to normal tissue. After Gilchrist et al. [114] proposed magnetic hyperthermia in 1957, more researchers turned their attention to magnetic hyperthermia. Magnetic hyperthermia implants magnetic nanomaterials into tumor tissue and increases the local temperature of tumor tissue through the action of external alternating magnetic fields, which can be used to treat deep tumors. Many studies have proven the targeting of magnetic hyperthermia, which is different from traditional thermal therapy and has been greatly improved and enhanced [115,116,117].

Currently, nanoparticles with appropriate particle size, good biocompatibility, and low toxicity can be synthesized for targeted delivery of chemotherapy drugs, magnetic hyperthermia, and tumor immunity, showing great application potential in tumor treatment. In recent years, significant breakthroughs have been achieved in the precision diagnosis and treatment of various cancers, including breast cancer, liver cancer, and ovarian cancer [118,119,120].

In the field of breast cancer, researchers developed a magnetic biointerface (MPMB) platform targeting EpCAM/HER2: using magnetic nanoparticles as a substrate, modified with a polydopamine (PDA) coating, and grafted with dual-targeted peptides (EpCAM/HER2) and a bovine serum albumin (BSA) anti-fouling layer to achieve efficient capture of circulating tumor cells (CTCs). Clinical testing demonstrated a 70% positive detection rate for CTCs in breast cancer patients, significantly outperforming traditional single-target capture methods. It also reduces sample requirements and shortens detection time, providing an efficient tool for early breast cancer diagnosis and treatment monitoring. In liver cancer diagnosis and treatment, a novel magnetic nanoprobe exhibits outstanding precision imaging capabilities. Composed of disulfide-linked ultrafine iron oxide nanoparticles, this probe intelligently converts T2/T1 MRI signals by recognizing glutathione concentration differences between tumors and normal liver tissue. Its detection sensitivity for millimeter-sized liver cancer lesions significantly surpasses that of the clinically prevalent gadopentetate dimeglumine (Gd-DTPA) contrast agent, effectively addressing the diagnostic limitations in early-stage liver cancer. For ovarian cancer, cobalt-doped iron oxide nanoparticles achieve highly efficient magnetothermolysis through their unique cubic double-cone structure. Their heating rate under an alternating magnetic field reaches 3.73 °C/s, twice that of conventional magnetic nanoparticles. Combined with cancer-targeting peptide-mediated tumor enrichment, intravenous injection followed by 30 min of non-invasive magnetic field therapy can inhibit ovarian cancer tumor growth in mice, while the low dose requirement reduces systemic toxicity [121].

In summary, through precise targeting modifications, signal-responsive design, and multimodal functional integration, magnetic nanoparticles demonstrate significant application potential in the early diagnosis, targeted therapy, and efficacy evaluation of various cancers, providing crucial technical support for precision tumor diagnosis and treatment [122].

#### 2.2.4. Novel Metal Nanoparticles

In recent years, beyond traditional gold, silver, and iron-based nanoparticles, novel metal/metalloid nanoparticles represented by selenium (Se), tellurium (Te), and bismuth (Bi) have emerged as research hotspots due to their combined therapeutic activity and diagnostic capabilities. These materials not only exert direct antitumor effects through their physicochemical properties but also serve as drug carriers or diagnostic/therapeutic probes, enabling integrated “diagnosis-treatment-monitoring” and offering novel approaches for precision cancer therapy. Selenium, an essential trace element in the human body, participates in the synthesis of antioxidant enzymes like glutathione peroxidase, and its compounds exhibit inherent biocompatibility and low toxicity.

In recent years, selenium nanoparticles (SeNPs) have emerged as novel research subjects in tumor therapy due to their controllable size, tunable surface properties, and multifunctional biological activities.

SeNPs exert cytotoxic effects by “bidirectionally regulating” oxidative stress levels in the tumor microenvironment: at high concentrations, they promote excessive accumulation of reactive oxygen species (ROS) within tumor cells, disrupting mitochondrial membrane potential and activating the caspase-3/9 apoptosis pathway to induce apoptosis in various tumor cells including hepatocellular carcinoma, breast cancer, and colorectal cancer; At low concentrations, they scavenge excess ROS in normal tissues, mitigating oxidative damage induced by chemotherapy or radiotherapy to achieve “selective killing” [123,124]. For instance, studies reveal that PEG-modified SeNPs significantly inhibit HepG2 hepatocellular carcinoma proliferation by downregulating Bcl-2 (anti-apoptotic protein) and upregulating Bax (pro-apoptotic protein) expression, while exhibiting only one-fifth the toxicity to normal hepatocytes (L02) compared to the traditional chemotherapeutic agent doxorubicin [125,126]. Furthermore, SeNPs can suppress the NF-κB inflammatory pathway, reduce the secretion of pro-inflammatory factors such as IL-6 and TNF-α in the tumor microenvironment, and weaken the invasive and metastatic capabilities of tumor cells [125].

SeNPs surfaces are rich in active groups such as hydroxyl and amino groups, facilitating the loading of hydrophobic chemotherapeutic drugs like paclitaxel and curcumin through covalent or non-covalent interactions. This approach addresses issues of low solubility and poor bioavailability. Simultaneously, surface modification with targeting ligands such as RGD peptides, folic acid, and hyaluronic acid enables precise drug delivery to tumor sites [127]. For instance, Ahmad et al. [128] demonstrated that folate modification significantly enhances the tumor targeting of selenium nanoparticles. These nanoparticles themselves exhibit high efficacy and low toxicity in breast cancer treatment through multiple mechanisms, inducing oxidative stress, arresting cell cycles, modulating immunity, and inhibiting angiogenesis, providing experimental evidence for the application of selenium-based nanomaterials in precision cancer therapy. Furthermore, SeNPs exhibit excellent photothermal conversion properties. Under near-infrared (NIR) irradiation, they generate localized high temperatures that not only directly thermally ablate tumors but also trigger carrier disintegration, enabling “light-controlled release” of drugs and further enhancing local drug concentration [129,130].

While both tellurium and selenium belong to the oxygen group elements, tellurium nanoparticles (TeNPs) exhibit stronger surface plasmon resonance effects and photothermal conversion efficiency, making them ideal materials for PTT [131]. TeNPs rapidly convert light energy into heat under near-infrared irradiation, elevating local tumor temperatures to 42–45 °C to induce protein denaturation and necrosis in tumor cells. simultaneously, the generated thermal effect enhances tumor cell membrane permeability, promoting the endocytosis of chemotherapeutic drugs or immunomodulators, thereby achieving synergistic “photothermal-chemotherapy” or “photothermal-immunotherapy” treatment [132,133].

Although novel metal nanoparticles demonstrate multifunctionality and high efficacy in tumor therapy, their clinical translation still faces numerous challenges: First, the long-term in vivo metabolism and toxicity mechanisms of certain materials such as tellurium and bismuth remain incompletely understood, necessitating more systematic animal studies to evaluate their potential effects on the liver, kidneys, and hematopoietic system [134,135]; Second, during large-scale nanoparticle production, controlling size uniformity and surface modification stability proves difficult, potentially compromising batch-to-batch therapeutic consistency. Third, tumor microenvironment heterogeneity—such as variations in pH and H_2_O_2_ concentration—may cause instability in nanoparticle catalytic activity or drug release efficiency [136,137].

### 2.3. Carbon-Based Nanomaterials

#### 2.3.1. Carbon Nanotubes

Carbon nanotubes (CNTs) are hollow tubular inorganic nanomaterials formed by the curling of graphene sheets. Carbon atoms undergo sp^2^ hybridization to construct a two-dimensional hexagonal lattice. This unique structure endows the material with diverse mechanical and physicochemical properties. Based on the number of graphene layers, carbon nanotubes are categorized into single-walled carbon nanotubes (SWCNTs) and multi-walled carbon nanotubes (MWCNTs). CNTs can achieve efficient drug loading and sustained release due to their physicochemical properties, high aspect ratio, good electrical and mechanical properties, and physical adsorption on cell membranes. The tubular structure is easy to cross the membrane, and its properties are stable and easy to modify, so it has attracted widespread attention in the delivery research of anti-tumor drugs [138]. Beyond carbon nanotubes, cubic carbon nanostructures (CCNs) have emerged as a new member of the carbon nanomaterials family, garnering significant attention in recent years. Formed primarily by sp^3^-hybridized carbon atoms into cubic crystal structures like diamond and diamond-like phases, CCNs combine high hardness, outstanding chemical stability, and excellent biocompatibility. Their surfaces can be readily functionalized through modification with functional groups such as hydroxyl and carboxyl groups. They have demonstrated application potential in fields including drug delivery, bioimaging, and tumor therapy [139].

CNTs have been widely used as a new type of in vitro drug carrier. CNTs have demonstrated efficient intracellular delivery of diverse biomolecules, including drugs, peptides, proteins, plasmid DNA, and small interfering RNA (siRNA), primarily via endocytic pathways. Meanwhile, the inherent ability of carbon nanotubes to absorb near-infrared (NIR) light has been exploited for the in vitro destruction of cancer cells. Notably, unlike the tubular structure of carbon nanotubes and their sp^2^ hybridization-dominated photothermal properties, cubic nanocarbon exhibits unique advantages in drug loading due to its distinctive electronic structure and crystalline morphology. For instance, the porous structure of diamond-like cubic nanocarbon can load hydrophobic chemotherapeutic drugs via physical entrapment or covalent bonding, achieving loading capacities of 30–40% of its own mass. These drugs are slowly released in the acidic tumor microenvironment, prolonging their action time [140,141]. Ghosh et al. [142] demonstrated that paclitaxel (PTX) coupled to PEGylated SWCNTs and injected into mouse xenograft tumors had a higher tumor inhibition effect than the clinical paclitaxel drug formulation. SWCNTs exhibit strong optical absorption spanning from the ultraviolet to NIR regions, and the heat produced through NIR absorption can be effectively utilized in photothermal therapy. Jeyamohan et al. [143] constructed a DOX-FA-PEG-SWCNTs composite material, which further enhanced its ability to kill tumor cells by utilizing the photothermal effect of SWCNTs and targeted tumor destruction mediated by NIR radiation. In contrast, cubic nanocarbon exhibits slightly lower light-to-heat conversion efficiency than carbon nanotubes. However, its superior chemical stability and lower biotoxicity confer greater potential for long-term in vivo drug delivery systems. For instance, hydroxyl-modified cubic nanocarbon loaded with siRNA effectively circumvents degradation by nucleases in vivo, thereby enhancing siRNA bioavailability [144].

At present, most experiments on carbon nanotube composite materials are still in the stage of in vitro and animal experiments. Research on cubic nanocarbon has primarily focused on material synthesis and in vitro functional validation. Issues such as its long-term metabolic patterns in vivo and the crystallographic uniformity of large-scale preparations remain unresolved, and more clinical applications and practical support are needed. With the continuous development of science and technology and the deepening of research, carbon nanotubes, as a nanomaterial with excellent performance, will show greater potential in the application of biomedicine. A human-friendly nanofunctional material with good biocompatibility, safety, and non-toxicity is expected to be born shortly.

#### 2.3.2. Graphene Oxide

Graphene is a two-dimensional sheet-like material composed of carbon atoms arranged in a hexagonal aromatic ring structure through sp^2^ hybridization. The individual layers are held together by van der Waals forces and π–π interactions, forming a stacked architecture [145]. In monolayer graphene, each carbon atom has four valence electrons: three participate in forming σ bonds with adjacent carbon atoms, while the remaining unpaired electron contributes to delocalized π bonds. These delocalized π-electrons are free to move across the entire graphene sheet, endowing the material with exceptional electrical conductivity and outstanding photothermal conversion capabilities [146]. Owing to its unique electronic structure and physicochemical properties, graphene has emerged as a key focus in materials research in recent years [147,148].

Nanoparticles of graphene oxide (NGO), a derivative of graphene, are produced through oxidation using strong oxidizing agents followed by ultrasonic fragmentation or mechanical exfoliation, resulting in the introduction of abundant oxygen-containing functional groups such as carbonyl, hydroxyl, carboxyl, and epoxy groups [149,150]. Compared to conventional large-sized graphene sheets, NGO exhibits superior water solubility, higher surface activity, and more abundant reactive sites, leading to enhanced electrical, optical, mechanical, and photothermal properties [151]. These characteristics not only broaden its potential in fields such as energy storage, sensing, and quantum devices, but also lay a solid foundation for its biomedical applications, particularly in cancer therapy [152].

In recent years, NGO have attracted increasing attention in cancer treatment, especially as a drug delivery platform, where they demonstrate remarkable advantages. The abundant oxygen functional groups on its surface facilitate chemical modification and efficient conjugation with a variety of drugs, targeting ligands, and biomacromolecules, making it an ideal carrier for hydrophobic anticancer agents [153]. Furthermore, the excellent photothermal performance of NGO enables its use as an effective photosensitizer. As a central platform, NGO have been employed to integrate chemotherapy, PTT, and immunotherapy, leading to the development of multimodal and multifunctional therapeutic strategies for comprehensive cancer treatment [154]. As an example, the experiment conducted by Yang et al. [155] designed a multifunctional nanotherapeutic system based on the photothermal properties of NGO, which targets tumor cells, inhibits epidermal growth factor receptor signaling, and combines PTT with chemotherapy.

NGO-based photothermal therapy possesses several unique advantages for clinical translation, including remote activation, spatial and temporal selectivity, high reproducibility, and relatively low tissue toxicity [156]. More importantly, NGO platforms can be engineered into multifunctional nanocarriers capable of generating heat under NIR laser irradiation while simultaneously delivering chemotherapeutic agents, immunomodulators, or photosensitizers. This enables targeted delivery and localized treatment of tumor tissues [157]. Such multifunctional systems not only enhance therapeutic targeting and efficacy but also reduce systemic side effects associated with chemotherapy and immunotherapy. By increasing local drug concentration within tumor tissues, they improve treatment safety and prognosis [158].

In summary, the commonly used nanomaterials in tumor therapy, including metal, polymer, lipid-based, and carbon-based materials, each emphasize distinct structural compositions, core characteristics, and functional roles (Table 1). However, they all revolve around the central objective of enhancing the precision of tumor treatment while reducing systemic toxicity.

The commonality among nanomaterials lies in their ability to overcome the limitations of traditional cancer treatments—insufficient efficacy, excessive toxicity, and low delivery efficiency—through size regulation, surface functionalization, and structural optimization. In the future, as advancements are made in material synthesis techniques, targeting strategies, and clinical translation research, the performance of various nanomaterials will be further optimized. Their application in cancer treatment will evolve from single-therapy delivery toward an integrated “diagnosis-treatment-monitoring” approach, providing more efficient solutions for tackling complex oncological challenges.

## 3. Application of Nanomaterials in Tumor Treatment

As an emerging multifunctional drug delivery platform, nanomaterials offer several advantages over traditional drug carriers. They enable targeted delivery, controlled release, and prolonged retention of therapeutic agents at disease sites, thereby significantly enhancing drug bioavailability and therapeutic efficacy while minimizing off-target toxicity and adverse effects on healthy tissues [175]. In addition to serving as carriers for chemotherapeutic agents, certain nanomaterials possess intrinsic physicochemical properties—such as photothermal conversion capability and photosensitivity—that allow them to actively participate in therapeutic processes [176].

In cancer treatment, nanomaterials have evolved from simple drug delivery vehicles into integrated therapeutic platforms with multiple functionalities. Their high photothermal conversion efficiency can be utilized in PTT, where externally applied laser irradiation induces localized hyperthermia to ablate tumor cells [177]. Specific types of nanomaterials can also generate ROS upon activation, enabling their application in PDT [178]. With the advancement of research on the tumor immune microenvironment, nanomaterials have been increasingly employed to deliver immunostimulants, antibodies, or nucleic acid-based therapeutics for precise regulation of cancer immunotherapy [179].

Owing to their high programmability and capacity for multifunctional integration, nanomaterials have facilitated the development of multimodal combination therapy systems. These platforms synergistically integrate chemotherapy, PTT, PDT, and immunotherapy to enhance therapeutic outcomes, overcome tumor heterogeneity, and address multidrug resistance [180,181,182]. Such advanced therapeutic strategies are gradually shaping the future direction of cancer treatment.

The following sections will provide a comprehensive overview of the multiple applications of nanomaterials in cancer therapy, focusing on five key aspects: targeted drug delivery systems, photothermal therapy, photodynamic therapy, nano-immunotherapy, and multimodal combination therapy. This review aims to highlight recent advances and future prospects of nanomaterial-based strategies for precise, efficient, and low-toxicity cancer treatment.

### 3.1. Targeted Drug Delivery System

Targeted drug delivery systems (TDDS) use the unique properties of nanomaterials to deliver therapeutic drugs directly to the tumor site, thereby improving efficacy and reducing side effects. In TDDS, the typical nanoparticle size ranges from approximately 10 to 200 nanometers. Nanoparticles within this size range can utilize the EPR effect to passively accumulate in tumor tissues, and their relatively large size helps prevent swift elimination through renal filtration [183]. Moreover, the physicochemical characteristics of nanoparticles—including their size, morphology, and surface charge—can be finely tuned by altering the material composition and adjusting component ratios. TDDS operate via two primary mechanisms: passive targeting, which relies on the EPR effect, and active targeting, mediated by specific ligand–receptor interactions [184].

Nanocarriers commonly used in TDDS include liposomes, polymer nanoparticles, dendrimers, carbon-based nanomaterials, and inorganic nanoparticles. These carriers can be engineered to improve their stability, drug loading capacity, and release kinetics [185]. By modifying the surface of nanomaterials, such as using antibodies or peptide targeting ligands or pH/enzyme-responsive groups, drugs can be precisely delivered to tumor sites to reduce damage to normal tissues. Nathalie et al. found that combining magnetic targeting can significantly increase tumor accumulation. The PLGA-SPIO nanoparticles were targeted by RGD and magnetic fields, and the tumor uptake was increased by about 8 times compared with passive targeting [186].

The microenvironment of a tumor is different from that of normal healthy tissue in the body. One obvious difference is the pH. The overall pH of the human body is alkaline at 7.4, while the pH of tumor tissue ranges from 7.0 to 7.2. pH-responsive tumor-targeted drug delivery is a special form of targeted drug delivery that uses nanoparticles to deliver therapeutic drugs directly to cancerous tumor tissue while minimizing their interaction with healthy tissue [187]. Advances in smart drug release systems. Stimuli-responsive nanocarriers that respond to pH, temperature, redox state, or enzymatic reactions in the tumor microenvironment have shown the potential to improve therapeutic precision [188]. These drug delivery systems are designed to respond to the pH environment of diseased or cancerous tissues, triggering structural and chemical changes within the drug delivery system [189]. This form of targeted drug delivery is localized, prolongs drug action, and protects the drug from being broken down or eliminated by the body before it reaches the tumor.

Nanomaterials can be used as carriers in tumor-targeted drug delivery systems to achieve specific enrichment and responsive release of drugs at the tumor site, thereby effectively improving efficacy and reducing damage to normal tissues [190]. Due to their suitable particle size, they have an ultra-high surface area to volume ratio. This enables them to efficiently load drugs and achieve passive targeted accumulation around tumors due to their leaky blood vessels (EPR effect) [191]. In addition, the high surface-to-volume ratio of nanoparticles allows for a variety of chemical modifications, including stealth coatings and targeting ligands. For instance, modifying the surface with PEG can prolong circulation time by minimizing photodegradation and decreasing clearance mediated by the mononuclear phagocyte system (MPS) [34]. These features together enhance the pharmacokinetics and biodistribution of anticancer drugs.

In addition to passive targeting, nanocarriers can be actively directed to tumor cells through surface binding to ligands such as folate, antibodies, or tumor-homing peptides [192]. For example, folate-functionalized nanoparticles target folate receptors, which are overexpressed on various tumor types, promoting receptor-mediated endocytosis. Responsive nanocarriers provide additional specificity by releasing their payload in response to stimuli characteristic of the tumor microenvironment. pH-sensitive materials remain stable in the blood (pH ~7.4) but break down under the acidic conditions of tumor tissue or endosomes (pH < 6.5), triggering controlled drug release [193]. Similarly, enzyme-responsive systems utilize cleavable linkers that are sensitive to matrix metalloproteinases or hyaluronidase, which are overexpressed in tumors [194].

Encapsulating drugs within nanocarriers provides several therapeutic benefits, such as enhancing the solubility of hydrophobic agents like paclitaxel, preventing early degradation, and minimizing systemic toxicity. Approved nanomedicines such as Doxil^®^ (liposomal doxorubicin) and Abraxane^®^ (albumin-bound paclitaxel) exemplify the clinical translation of these benefits [48]. Controlled-release properties further enhance therapeutic efficacy, ensuring sustained drug exposure at the tumor site while minimizing off-target effects. Recent designs incorporate magnetic, photothermal, or ultrasound triggering systems, providing additional layers of spatial and temporal control [195].

Passive targeting takes advantage of the EPR effect characteristic of tumor tissues, which is characteristic of solid tumors because of their abnormal blood vessel architecture and compromised lymphatic drainage. This phenomenon allows nanoparticles to accumulate preferentially in tumor tissues [196]. In contrast, active targeting enhances cellular uptake through ligand-receptor interactions. Although relying on initial passive accumulation, active targeting enhances internalization by cancer cells and improves the therapeutic index. Combination strategies that exploit these two mechanisms are gaining popularity in modern nanocarrier design [197]. In clinical applications, folate-modified liposomal delivery systems demonstrate precision targeting potential in ovarian cancer therapy. A Phase I clinical trial evaluated the safety and efficacy of folate receptor-targeted liposomes loaded with doxorubicin. This formulation achieves active targeted delivery by leveraging the specific binding of folate to folate receptors highly expressed on tumor cells. The trial has now advanced to Phase II, focusing on assessing therapeutic efficacy in platinum-resistant ovarian cancer [198].

Traditional drug delivery systems have some limitations in terms of drug distribution, pharmacokinetics, in vivo degradation, drug solubility, and the ability to penetrate biological systems [199]. Conventional chemotherapy drugs are indiscriminately delivered throughout the body after administration, and the treatment is not targeted, with limited development prospects. In recent years, the field of nanotherapy has developed rapidly, and drug release and drug distribution monitoring can be completed simultaneously by modifying a single nanocarrier [200]. For instance, traditional chemotherapy drugs such as doxorubicin, paclitaxel, and cisplatin are widely used in clinical practice, but due to their low selectivity and high systemic toxicity, they often lead to serious adverse reactions [34]. Nanomaterials, as emerging drug delivery carriers, can achieve controlled release and precise positioning of chemical drugs by adjusting particle size, surface properties, and response mechanisms, thereby effectively improving the therapeutic index [201]. Nanoparticles can encapsulate a variety of small-molecule chemotherapy drugs. Liposomes and polymer nanoparticles are particularly effective for loading hydrophobic drugs. For instance, Abraxane^®^ is paclitaxel loaded into albumin nanoparticles, which successfully circumvents the toxicity issues associated with the Cremophor EL solvent.

### 3.2. Photothermal Therapy and Photodynamic Therapy

PTT and PDT are two rapidly evolving non-invasive strategies for cancer treatment, both of which rely on nanomaterials as core therapeutic mediators. PTT utilizes nanomaterials to convert NIR light into thermal energy, inducing localized hyperthermia that triggers apoptosis or necrosis of tumor cells. In contrast, PDT employs photosensitizers that, upon light activation, generate ROS, thereby causing oxidative stress and selectively killing cancer cells [202].

In PTT, commonly used photothermal agents include gold nanoparticles, carbon nanotubes, graphene oxide, and transition metal dichalcogenides (TMDs), all of which exhibit high photothermal conversion efficiency and favorable biocompatibility. In comparison, PDT employs a variety of photosensitizers, such as porphyrins, phthalocyanines, and novel nanocomposites, characterized by high ROS generation efficiency and tumor-targeting capability, thereby enhancing the selectivity and safety of treatment [203]. Liposomal nanosystems loaded with photosensitizers achieve a breakthrough in esophageal cancer treatment. In Phase I clinical trials, heme porphyrin monomethyl ether liposomes combined with 630 nm laser therapy enhanced photosensitizer accumulation at tumor sites via the EPR effect. The ROS generated by PDT significantly induced cancer cell apoptosis while causing minimal damage to normal mucosal tissue. This formulation has now advanced to Phase II clinical trials, focusing on exploring its combined application with radiotherapy [204,205].

Recent preclinical and clinical studies have demonstrated that combining PTT and PDT can produce significant synergistic therapeutic effects. Multifunctional nanoplatforms have been developed to enable this combination therapy, which not only enhances the destruction of tumor tissues but also reduces recurrence and systemic toxicity [206]. This synergism is primarily attributed to increased vascular permeability induced by PTT and immune activation triggered by PDT, resulting in a more comprehensive tumor eradication.

As a promising non-invasive treatment modality, PTT relies on photothermal agents to convert NIR laser energy into localized heat, enabling precise thermal ablation of cancer cells at the target site. Compared to traditional therapies, PTT offers several advantages, including minimal invasiveness, low toxicity, and reduced likelihood of drug resistance [207]. PDT, on the other hand, takes advantage of the elevated metabolic activity of tumor tissues, allowing preferential accumulation of photosensitizers at tumor sites. Upon irradiation with light of a specific wavelength, the excited-state photosensitizers transfer energy to surrounding oxygen molecules, generating ROS that selectively induce apoptosis in tumor cells [208].

Despite the substantial progress in the development of light-based therapeutic agents and optical devices, several key limitations persist. In PTT, some photothermal materials lack sufficient tumor-targeting capabilities, limiting therapeutic efficacy. In PDT, non-specific distribution of photosensitizers may lead to phototoxic side effects, thereby hindering clinical application [8]. To address these challenges, researchers are actively exploring strategies such as targeted modifications and tumor microenvironment-responsive systems to improve treatment specificity. 

Overall, PTT and PDT based on nanomaterials have become research hotspots in cancer therapy. In the future, the integration of drug delivery, photothermal effects, and immune activation into multifunctional nanocarriers holds great promise for improving therapeutic outcomes and enabling precise combination therapies for various types of cancer.

### 3.3. Nano-Immunotherapy

Tumor immunotherapy is an important direction of current tumor treatment by activating the body’s immune system to fight tumors. The development of nanotechnology has provided a new strategy for tumor immunotherapy, and nanomaterials have shown unique advantages in regulating the tumor immune microenvironment and enhancing immune responses. Immunotherapy, especially immune checkpoint inhibitors (ICIs) and tumor vaccines, has achieved remarkable results in the treatment of melanoma, non-small cell lung cancer, among others. in recent years [209]. However, due to the complexity of TME, the nonspecific distribution of drugs in the body, and the immunosuppressive state, immunotherapy alone still faces many challenges. To this end, researchers explored the combination of nanotechnology and immunotherapy to develop a new type of treatment strategy-nanoimmunotherapy [210]. Nanoimmunotherapy regulates the body’s immune system through nanotechnology, which can effectively overcome the problems of traditional immunotherapy, such as low bioavailability, poor targeting, and immunosuppressive microenvironments [211].

Nanocarriers can deliver antigens, adjuvants, antibodies, mRNA, and other immunologically active substances to immune organs or immune cells by encapsulation, adsorption, or covalent linkage, thereby increasing their concentration and biological stability in the target area. As an example, the experiment conducted by Hamouda et al. [212] enhanced the anti-tumor effect of CD8^+^ T cells in the tumor microenvironment by locally delivering mRNA expressing IL-21, IL-7, and 4-1 BBL, and induced a systemic, long-term tumor-specific immune memory response by activating dendritic cells and tumor-draining lymph nodes, thereby achieving efficient, safe, and lasting tumor clearance. This mechanism induces a large number of functional CD8^+^ T cells to infiltrate the tumor and produce high levels of granzyme B, IFN-γ, and TNF-α, significantly enhancing the ability to kill tumor cells.

Nanomaterials such as liposomes, polymeric nanoparticles, and metal–organic frameworks (MOFs) can significantly prolong their in vivo circulation half-life by constructing protective nanoencapsulation structures. Meanwhile, they can avoid uptake by non-target tissues and improve bioavailability [213]. Following the successful application of lipid nanoparticle technology in novel COVID-19 vaccines, this technology has been extended to the delivery of anti-tumor mRNA vaccines [214]. TME exhibits multiple immunosuppressive features, including hypoxia, acidic pH, elevated lactate levels, and the accumulation of regulatory T cells and M2-polarized macrophages, which collectively impair the infiltration and effector functions of cytotoxic T lymphocytes. Some “functionally responsive” nanomaterials can release active factors in response to specific stimuli in TME to achieve local regulation. Jiang et al. [215] proposed an innovative silicon-based oxygen-driven nanomotor (Si-motor) system for active enhancement of synergistic oxidative stress therapy in tumor areas. MnO_2_ nanoparticles can react with H_2_O_2_ in TME to produce oxygen, relieve hypoxia, and induce polarization of M1 macrophages to enhance anti-tumor immunity.

Nanoplatforms enable the coordinated delivery of multiple immunomodulatory agents, such as antigen–adjuvant combinations or anti-PD-L1 antibodies co-administered with TLR agonists, thereby enhancing T cell responses and promoting robust memory immune responses. As an example, the experiment conducted by Ugur et al. [216] used a lipid-polymer hybrid nanoplatform that can carry OVA antigen and CpG oligonucleotide (TLR9 agonist) at the same time, inducing persistent CD8^+^ T cell response and delaying tumor progression in a mouse tumor model.

Nano-immunotherapy, as a new strategy integrating nanotechnology and immune regulation mechanisms, has shown strong application potential in various tumor treatment modes. Its main application directions include combined immune checkpoint inhibition therapy, construction of nano-vaccines, and delivery of immune adjuvants, among others. ICIs such as anti-PD-1/PD-L1 and CTLA-4 antibodies are currently the core drugs for tumor immunotherapy, but their clinical response rates are still limited by insufficient tumor antigen presentation and the inhibitory environment of the TME. Nanocarriers can effectively increase the enrichment of immune checkpoint inhibitors in tumor sites, thereby enhancing the therapeutic effect and reducing immune-related side effects [217]. Nanoplatforms can enhance the efficacy of ICIs by targeted delivery of immune agonists to “heat” cold tumors [218], alleviate the immunosuppressive microenvironment [219], and locally controlled release [220].

Nanovaccines refer to the use of nanocarriers to deliver tumor antigens and immune adjuvants to antigen-presenting cells, enhancing dendritic cell activation and T cell response [221]. Using polymer nanoparticles as carriers for DNA vaccines can be an effective treatment method to activate immune responses and improve antigen presentation efficiency [222]. Peptide or fusion protein antigens can be simultaneously encapsulated in polymer nanoparticles, which helps to cope with tumor antigen heterogeneity and improve the broad spectrum of treatment [223].

Nano-immunotherapy has advantages over traditional therapies in terms of drug delivery efficiency, immune activation intensity, microenvironment regulation, and T cell function recovery. By precisely designing the composition, surface modification, and response mechanism of nano-carriers, more personalized and efficient solutions can be provided for tumor treatment.

### 3.4. Multimodal Combined Therapy

Multimodal combination therapy refers to the simultaneous or synergistic use of two or more therapeutic approaches to act on tumors, thereby producing a synergistic effect, improving the efficacy, and reducing the toxic side effects of a single treatment regimen, which may lead to dazzling super-additive therapeutic effects. For instance, certain anticancer agents possessing radiosensitizing properties not only exert cytotoxic effects through chemotherapy but also significantly enhance the susceptibility of tumor cells to ionizing radiation, thereby improving the efficacy of radiotherapy [224,225,226], which produces a stronger therapeutic effect than the theoretical combination of the corresponding individual treatments [227,228,229]. Compared with the limited therapeutic effects and possible side effects of monotherapy, the development of multimodal synergistic therapies may have the collective advantages of each treatment and produce higher anticancer effects while using lower doses of therapeutic drugs, thus avoiding the side effects caused by high doses [230].

The successful implementation of multimodal synergistic therapy depends primarily on the precise integration of various therapeutic modalities within a unified platform, rather than merely combining them to achieve enhanced synergistic effects. The rapid development of nanotechnology has made it possible to assemble multiple types of therapeutic agents into a nanostructure through physical adsorption and chemical binding forces, thereby generating multifunctional nanomaterials to achieve the “mission” of multimodal synergistic therapy [231].

In biomedical applications, nanomaterials demonstrate multiple significant advantages compared to traditional small-molecule probes. First, by leveraging the EPR effect, nanomaterials can passively accumulate and preferentially localize within tumor tissues [232]. Second, due to the rich presence of surface functional groups, nanomaterials can be easily functionalized with proteins, peptides, or other biomolecules, thereby reducing nonspecific recognition and clearance by the reticuloendothelial system (RES) [233,234]. Third, Nanomaterials possess a high surface area-to-volume ratio, which enables them to efficiently load substantial amounts of drugs, genes, and other therapeutic agents while shielding these cargos from enzymatic degradation within complex physiological environments [235]. Furthermore, functionalized nanomaterials are capable of controlling the release of encapsulated drugs in response to various internal and external stimuli, such as pH changes, glutathione levels, and light exposure, which effectively prevents premature drug leakage in healthy tissues and minimizes side effects [236,237].

In recent years, multimodal combination therapy has become a key strategy to improve anti-tumor efficiency. Nanomaterials have become an ideal platform for multimodal therapy due to their programmability, targeting, and multifunctional loading capacity [238].

Multimodal combination therapy produces stronger anti-tumor effects than single therapy through synergistic effects. The main principles include mechanism complementarity and target complementarity. When multiple treatment methods are carried out simultaneously, tumor cells can be attacked through multiple signal pathways at the same time, increasing their difficulty in escaping or drug resistance [239]; rationally designed nanoplatforms can achieve spatial site stratification and improve the integrity of tumor clearance [240] (Figure 5); nanomaterials can be programmed to achieve sequential release of multiple components and optimize treatment timing [241]; multimodal therapy can attack tumors from multiple angles and avoid drug resistance caused by single-drug or single-pathway treatment [242].

As a programmable and multifunctional therapeutic platform, nanomaterials offer a high degree of integration and precise regulation, making them ideal carriers for multimodal synergistic cancer therapy. By incorporating multiple therapeutic modalities into a single nanoplatform, these systems can simultaneously target various biological pathways within tumors and elicit synergistic effects that surpass the sum of individual treatments. Studies have demonstrated that combining multiple therapeutic strategies within a single nanomaterial-based platform often yields superior overall efficacy compared to the additive effects of each modality alone [243].

Currently, nanomaterial-enabled multimodal combination therapies include bimodal and trimodal treatment strategies, both of which have been shown to produce enhanced anticancer outcomes [244]. Common combinations include photothermal–chemotherapy, photothermal–photodynamic therapy, magnetothermal–chemotherapy, and immunotherapy–photothermal therapy. These integrated approaches offer promising solutions to major challenges in cancer treatment, such as tumor heterogeneity, multidrug resistance, and immune evasion.

In multimodal cancer treatment systems, the synergistic combination of PTT and chemotherapy represents one of the most extensively studied therapeutic modalities. PTT relies on the ability of nanomaterials to absorb NIR light and convert it into heat, resulting in localized hyperthermia within the irradiated region to induce tumor cell damage and apoptosis. However, the therapeutic efficacy of PTT may be limited in non-irradiated tumor tissues or deeply seated cancer cells, as the thermal effect is often confined to the area exposed to light. To overcome this limitation and enhance both the spatial and temporal efficacy of treatment, researchers have increasingly explored the strategy of combining PTT with chemotherapy [245]. By incorporating chemotherapeutic agents into photothermal-responsive nanocarriers, drug release can be accelerated in response to localized heat generated under NIR irradiation. This simultaneous induction of hyperthermia and chemotherapeutic cytotoxicity within the tumor microenvironment significantly enhances the overall therapeutic efficacy, increases tumor cell killing, and reduces the likelihood of recurrence. This synergistic therapeutic approach has been widely applied in various solid tumor models, demonstrating promising therapeutic outcomes and translational potential. For instance, Zhang et al. [246] developed a polyethylene glycol-modified nano-graphene oxide system loaded with doxorubicin (NGO-PEG-DOX) for the combined application of chemotherapy and photothermal therapy within a single system. The functionalized graphene oxide enabled effective ablation of tumors both in vitro and in vivo. The NGO-PEG-DOX nanoparticles demonstrated a synergistic effect by integrating localized chemotherapy with externally triggered NIR photothermal therapy, thereby significantly improving anticancer efficacy compared to monotherapies.

The combination of PDT and PTT has also emerged as a potent strategy to improve cancer treatment outcomes. PDT is a light-activated process in which photosensitizers generate ROS upon light exposure to induce tumor cell death. While PDT is minimally invasive and has limited damage to normal tissues, its therapeutic efficacy is often compromised in the hypoxic tumor microenvironment. When combined with PTT, the resulting hyperthermia can relieve tumor hypoxia and augment PDT efficacy [247,248]. Various nanocarrier platforms have been developed for the co-delivery of PTT and PDT agents, including liposomes, micelles, nanoemulsions, proteins, and polymer-based nanoparticles. These carriers offer excellent biocompatibility, controlled degradability, and efficient drug-loading capacity, making them ideal candidates for dual phototherapeutic applications [249]. For example, Li et al. [250] developed a multifunctional nanoplatform integrating polypyrrole (PPy) and MnO_2_ for combined PTT and enhanced PDT, which resulted in a tumor cell survival rate as low as 1.29%, markedly lower than that observed with single therapies. Liu et al. [251] employed PEG-modified molybdenum disulfide (MoS_2_) nanosheets co-loaded with the photosensitizer chlorin e6 (Ce6), demonstrating effective synergistic PTT/PDT both in vitro and in vivo, with enhanced tumor cell ablation. Similarly, Guo et al. fabricated folic acid (FA)- and Ce6-functionalized graphene oxide for targeted PTT/PDT combination therapy, which exhibited enhanced cytotoxicity against various cancer cell types and showed promise for other diseases as well [252]. Through the integrative capabilities of nanoplatforms, simultaneous delivery and synchronized release of both photosensitizers and photothermal agents at tumor sites can be achieved. This approach significantly improves therapeutic efficacy, minimizes systemic toxicity, and offers new possibilities for clinical translation.

As a therapeutic strategy that utilizes magnetic nanoparticles to generate localized heat under an alternating magnetic field for tumor ablation, magnetic hyperthermia therapy (MHT) has demonstrated promising efficacy in the local control of tumors [253]. However, its effectiveness against metastatic tumors remains limited. To address this challenge, current research has increasingly focused on combining MHT with systemic therapies such as chemotherapy or immunotherapy to enhance drug or immune effector penetration within tumors and improve overall therapeutic outcomes [254]. For instance, Pan et al. [255] developed monodisperse, high-performance CoFe_2_O_4_@MnFe_2_O_4_ nanoparticles capable of inducing primary tumor ablation through magnetic hyperthermia while simultaneously releasing tumor-associated antigens to activate dendritic cells and cytotoxic T lymphocytes. This synergistically enhanced the efficacy of checkpoint blockade immunotherapy and inhibited metastatic tumors. Similarly, Liu et al. [256] designed clover-shaped magnetic nanoclusters (CSMNs) that achieved non-invasive MHT combined with systemic chemotherapy via tumor vasculature targeting, demonstrating significantly improved therapeutic outcomes in brain cancer models.

Although ICIs have shown remarkable efficacy in cancer therapy, their overall clinical response rate remains suboptimal. In recent years, nanomaterials have been extensively explored to enhance the efficacy of immunotherapy. They not only induce immunogenic cell death (ICD) to activate anti-tumor immune responses but also serve as efficient delivery systems for immune-modulatory molecules or RNA-based therapeutics. These strategies promote the transformation of “cold” tumors into “hot” tumor microenvironments, thereby increasing the responsiveness and therapeutic outcomes of immunotherapy [257,258].

Owing to their unique physicochemical properties and engineering flexibility, nanomaterials have demonstrated great potential in various critical aspects of cancer therapy. From targeted drug delivery to phototherapeutic modalities such as PTT and PDT, as well as modulation of the tumor immune microenvironment and the development of multimodal synergistic therapeutic systems, nanoplatforms are continuously propelling cancer treatment toward more personalized and precise strategies [259].

Benefiting from their tunable size, surface modifiability, and favorable biocompatibility, nanocarriers enable the precise delivery of anticancer agents, significantly enhancing therapeutic efficacy while minimizing systemic toxicity. In photothermal and photodynamic therapies, nanomaterials exhibit strong optical responsiveness, allowing efficient conversion of external energy into selective tumor cell ablation. Simultaneously, by inducing immunogenic cell death, delivering immunomodulatory agents, or transporting RNA-based therapeutics, nanoplatforms offer effective strategies for immunotherapy, facilitating the transformation of “cold” tumors into “hot” immune-responsive microenvironments. In multimodal treatment strategies, nanomaterials serve as integrated platforms that enable the co-delivery of chemotherapeutics, thermal agents, photosensitizers, and immunomodulators, thereby enhancing therapeutic efficacy and overcoming the limitations of monotherapies [260].

Currently, nanomaterial-mediated multimodal combination therapies have evolved into diverse approaches encompassing combinations of different mechanisms of action. These approaches have demonstrated significant advantages in preclinical studies, though variations exist among them in terms of applicable scenarios, technical bottlenecks, and clinical translation potential. To clearly contrast the core characteristics of each approach, the table below systematically summarizes mainstream nanomaterial-mediated multimodal combination therapies for tumors. It includes their principles, carrier selection, advantages and disadvantages, representative cases, and clinical translation prospects, providing a reference for research and application in this field (Table 2).

Despite substantial progress in preclinical research, the clinical translation of nanomaterials still faces multiple challenges, including production standardization, long-term safety evaluation, immune response regulation, and interpatient variability. To fully realize the potential of nanotechnology in precision oncology, further optimization of nanosystems should focus on improving therapeutic selectivity, verifying long-term biocompatibility, and achieving scalable and controlled synthesis. With the continuous advancement of interdisciplinary technologies and preclinical investigations, nanomaterials are expected to play a more prominent role in precision cancer treatment, accelerating the clinical translation of personalized therapeutic strategies [276].

## 4. Challenges and Issues of Nanomaterials in Tumor Therapy

One of the core applications of nanomaterials in tumor treatment is as a drug delivery system, which can achieve precise delivery of anti-tumor drugs. However, there are still significant challenges in the delivery efficiency and biodistribution of nanomedicines in the body, which seriously limit their clinical transformation potential. Nanomedicines have not yet been fully applied in the field of tumor treatment, and there are many challenges in research and development.

### 4.1. Drug Delivery and Biodistribution

Most nanomedicines rely on passive targeting mechanisms, especially enhanced permeation and EPR effects, to achieve selective aggregation through the high permeability of tumor blood vessels. However, studies have shown that the performance of the EPR effect in human tumors is far less ideal than that in animal models, and its spatial distribution is highly heterogeneous, resulting in uneven distribution of nanomaterials in tumor tissues and uneven therapeutic effects [137].

After entering the blood circulation, nanomaterials will quickly interact with plasma proteins to form the so-called “protein corona”, changing their original surface properties and affecting their recognition, uptake, and metabolism in the body [277]. This protein adsorption phenomenon often causes nanoparticles to be recognized and cleared by the MPS or the RES, thereby enriching non-targeted organs such as the liver and spleen, reducing the amount of drug accumulation in the tumor site [278].

The differences in the stability of different types of nanomaterials in the blood will also affect their pharmacokinetic behavior [279]. Notably, PLGA nanoparticles undergo rapid degradation in the bloodstream, leading to excessively fast drug release, whereas metal nanomaterials may aggregate as a result of nonspecific adsorption of surface functional groups, thereby exacerbating liver toxicity [280]. Some studies have attempted to improve delivery efficiency through active targeting modification, such as modifying the surface of nanoparticles with tumor-targeting ligands such as folic acid, HER2 antibodies, and RGD peptides to enhance binding to tumor cell surface receptors [281,282]. However, this active targeting strategy is also affected by protein corona interference and the dynamic blood flow environment in vivo, and the effect is often good in vitro but not as expected in vivo [36].

In recent years, increasing attention has been paid to the systematic and quantitative analysis of the in vivo biodistribution of nanomaterials. To address this need, researchers have increasingly employed multimodal strategies that integrate in vivo imaging techniques with fluorescence-based or fluorescence-quenching technologies, enabling dynamic monitoring of nanoparticle distribution, residence time, and clearance pathways across various tissues and organs. These efforts provide critical insights for the rational design and optimization of nanocarrier-based delivery systems [146]. Therefore, improving the accumulation efficiency of nanodrugs in tumor tissues, reducing their nonspecific enrichment in non-target organs, and enhancing their stability and durability in the body are the current key directions for optimizing drug delivery systems.

### 4.2. Toxicology and Biocompatibility

Although the application of nanomaterials in tumor treatment has advantages such as strong targeting and high penetration, their potential toxicological problems and lack of biocompatibility have always been one of the core obstacles restricting their clinical transformation. The in vivo behavior of nanomaterials is significantly different from that of traditional small-molecule drugs. Its long-term retention, nonspecific distribution, tissue accumulation, and interaction with host cells may trigger multiple biological reactions [283,284].

#### 4.2.1. Toxicity Induced by Particle Size and Surface Properties

The toxicity of nanomaterials mainly depends on their physical and chemical properties, such as particle size, morphology, charge, and surface modification. Studies have shown that nanoparticles with a diameter of less than 10 nm are more likely to be excreted from the body through the glomerular filtration system but are also more likely to penetrate cell membranes, enter organelles, and interfere with their functions, and may even cross the blood–brain barrier and cause neurotoxicity [285]. In addition, nanoparticles with positive charges on the surface are prone to strong interactions with cell membranes, leading to changes in membrane permeability and inducing cell apoptosis or necrosis [286].

#### 4.2.2. Oxidative Stress and Inflammatory Response

Some metal-based nanomaterials easily induce ROS production in cells, destroy mitochondrial function, and trigger DNA damage, lipid peroxidation, and inflammatory cascades [287,288]. This oxidative stress-related toxicity has been observed in a variety of in vivo models and may cause toxic manifestations such as pulmonary fibrosis and liver dysfunction.

#### 4.2.3. Immune Response and Immune Escape

Nanomaterials are recognized and engulfed by macrophages in the body, thereby activating the innate immune system, leading to the release of inflammatory cytokines, and triggering acute or chronic immune responses [289]. In particular, some unmodified inorganic nanomaterials are prone to inducing complement system activation and even triggering allergic reactions or “infusion reactions” [290]. In order to prolong the circulation time of nanoparticles in the body, researchers often use PEG coating strategies. However, in recent years, it has been found that some patients have anti-PEG antibodies in their bodies, which may lead to rapid clearance of drugs or even hypersensitivity reactions [291].

#### 4.2.4. Long-Term Residence and Chronic Toxicity

Non-degradable nanomaterials are slowly metabolized in the body and may be retained in the liver, spleen, bone marrow, and other organs for a long time, causing chronic inflammation, immunosuppression, or tissue structure damage. Animal experiments have shown that long-term low-dose intake of nanomaterials may also cause tissue lesions or tumor-like hyperplasia changes [292].

### 4.3. Scalable Production and Quality Consistency

As the research on nanomaterials in tumor treatment continues to deepen, the feasibility of their clinical transformation has attracted increasing attention. However, relative to conventional small-molecule drugs, the manufacturing of nanomedicines is more complex, requiring multiple components, numerous procedural steps, and strict control over various parameters [293].

Most nanodrugs are prepared in the laboratory by small-scale batch methods or solvent evaporation, self-assembly, emulsification, and other methods. These processes are extremely sensitive to environmental conditions. Slight changes can cause fluctuations in key parameters such as particle size, encapsulation efficiency, drug loading, and surface charge, which directly affect pharmacokinetics and therapeutic effects [294]. The particle size and surface charge of PLGA nanoparticles are influenced not only by the stirring rate but also by a range of physicochemical parameters, including solvent type, temperature, and pH. These factors become particularly critical during scale-up processes, where slight variations may lead to significant changes in nanoparticle characteristics [295].

With the rapid development of multifunctional nanomaterials, their structural design is becoming increasingly complex, involving the coordinated assembly of multiple functional components, such as magnetic cores, polymer shells, targeting ligands, and imaging probes. How to achieve the precise assembly and stable existence of these components under large-scale preparation conditions has become an important technical bottleneck restricting industrialization [296].

The key quality attributes of nanomedicines, such as particle size distribution, polydispersity index, Zeta potential, drug loading, and release curve, all require precise testing [293,297]. However, there is currently a lack of international, unified quality evaluation standards for nanomaterials. There are large differences in the detection methods and analysis results used by different laboratories, making it difficult to achieve regulatory uniformity [298]. For multi-component nanosystems, parameters such as stability, functional integrity, and interaction with biological systems need to be evaluated in a more systematic and multi-dimensional manner.

Traditional preparation processes mainly rely on intermittent manual operations, which are difficult to meet the industrialization requirements for “continuity, automation, and controllability”. In recent years, new processes such as microfluidics and continuous flow synthesis platforms have gradually been introduced into nano-drug preparation, which have good controllability and reproducibility, and can achieve particle size regulation and component homogenization [299,300].

### 4.4. Multidimensional Analysis of Key Bottlenecks in Nanomedicine Clinical Translation

The clinical translation process of nanomedicines is not solely dependent on ingenious material design and exceptional performance optimization. Instead, it is deeply entangled in a complex web of multidimensional obstacles, including regulatory approval, pharmacokinetic assessment, large-scale production, and cost control. We will conduct an in-depth analysis of these core bottlenecks, supplemented by case studies, offering significant reference value for advancing collaboration among industry, academia, research, and clinical application.

In the regulatory approval domain, major global regulatory bodies such as the FDA and the European Medicines Agency (EMA) adhere to a cautious “case-by-case assessment” principle for nanomedicines. To date, no unified guidance specifically targeting nanomedicines has been issued. This stems primarily from the unique physicochemical properties of nanomaterials—such as particle size and surface characteristics—which fundamentally differ from traditional drugs, necessitating the urgent development of targeted safety and efficacy assessment frameworks [301,302]. Taking the FDA approval pathway as an example, nanomedicines are typically submitted as either “New Molecular Entities (NMEs)” or “Biosimilars”, while additionally requiring submission of a comprehensive “nanomaterial assessment report”. This report must cover critical information, including particle size distribution, surface modification stability, potential for protein coating formation, and in vivo biodistribution data. Multifunctional nanosystems, such as “diagnostic-therapeutic” nanoparticles, face dual regulatory requirements for both drugs and medical devices. For instance, MRI contrast agent-chemotherapy conjugate nanoparticles must undergo rigorous validation of both imaging performance and therapeutic efficacy, significantly extending approval timelines by 30–50% compared to traditional drugs [303]. Compared to the FDA, the EMA imposes stricter approval standards for non-degradable nanomaterials like metallic nanoparticles, often requiring chronic toxicity data from animal studies spanning over 10 years. The case of BIND-014 exemplifies a classic regulatory approval failure. BIND-014, a polymeric nanoparticle targeting prostate-specific membrane antigen (PSMA) and loaded with docetaxel, demonstrated a 38% objective response rate in metastatic castration-resistant prostate cancer during Phase II clinical trials [304]. However, the FDA rejected its Phase III clinical trial application, citing “inability to confirm precise nanoparticle distribution within tumor tissue” and “insufficient long-term hepatotoxicity data.” The core issue lies in the drug’s failure to establish a quantitative relationship between “particle size distribution—hepatic clearance—toxicity”, thereby failing to meet the FDA’s stringent regulatory requirements for “predictability” in nanomedicines.

The pharmacokinetic behavior of nanomedicines differs significantly from that of traditional small-molecule drugs, necessitating the development of targeted assessment methods. The three key challenges lie in “distribution, metabolism, and clearance” [305]. Regarding distribution assessment, traditional pharmacokinetics primarily relies on plasma drug concentration monitoring. However, the pharmacokinetic behavior of “free drug” versus “nanoparticle-bound drug” in nanomedicines exhibits significant differences [304]. Take liposomal doxorubicin (Doxil) as an example: its plasma-free doxorubicin concentration accounts for only 5% of the total concentration, yet the free fraction is the primary form responsible for toxicity. Therefore, simultaneous detection of both the “nanoparticle carrier” and the “released drug” concentrations become an essential requirement. Regarding metabolism and clearance, degradable nanomaterials such as PLGA and liposomes require a comprehensive evaluation of their metabolites [305,306]. For instance, gold nanoparticles are primarily cleared via the MPS or RES. In such cases, close monitoring of phagocytosis by hepatic macrophages and spleen accumulation is essential to prevent long-term immunosuppression [307].

The transition from small-scale laboratory preparation to industrial-scale production presents a significant technological gap, which stands as one of the core obstacles in nanomedicine translation. This challenge manifests primarily in two critical aspects: “process controllability” and “scale-up effects” [308,309]. In laboratories, the commonly used “self-assembly method” heavily relies on manual operations, making it difficult to precisely control the uniformity of critical parameters such as temperature and stirring rate [310,311]. In contrast, industrial production urgently requires advanced technologies like continuous flow synthesis and microfluidics. Novartis employs microfluidic technology to manufacture the siRNA nanomedicine Patisiran. By precisely controlling fluid flow rates via computer, they achieved an outstanding particle size coefficient of variation (CV) below 5%, far surpassing traditional batch methods (CV > 20%) [299].

The production cost of nanomedicines is significantly higher than that of traditional drugs, primarily due to factors such as “raw material costs”, “process complexity”, and “quality control expenses”, severely limiting their widespread clinical adoption [312]. In terms of clinical cost-effectiveness, nanoparticle drugs typically command prices 5–10 times higher than conventional drugs [312].

The clinical translation of nanoparticle drugs constitutes a multidisciplinary systems engineering challenge, requiring simultaneous breakthroughs across the entire value chain: material design, pharmacokinetic evaluation, manufacturing processes, regulatory approval, and cost control. Future research should focus on: establishing internationally unified quality standards for nanomaterials, developing scalable production technologies such as continuous flow and microfluidics, and optimizing the balance between “targeting efficacy, biocompatibility, and cost.” Through collaborative efforts among industry, academia, research institutions, and clinical practitioners, nanomedicines hold promise to transition from “laboratory concepts” into routine clinical tools for cancer treatment.

## 5. Future Development and Prospects of Nanomaterials in Tumor Treatment

### 5.1. Safe and Controllable Material Design Strategy

The safety of nanomaterials has always been a core issue in their clinical transformation. For this reason, the strategy of “designing for safety from the source” has gradually become a research hotspot, which emphasizes the integration of controllable degradation, low immunogenicity, and low toxicity into the material synthesis stage. For instance, due to tumor heterogeneity, variations in hypoxia levels or the presence of particular enzymes essential for drug release may differ across metastatic sites, leading to unpredictable drug release profiles. A possible solution to improve tumor specificity is to use dual stimulus-responsive triggers [313,314,315], but special attention must be paid to further characterize these systems and improve the scalability of the formulation; for drug-loaded nanoparticles, optimized proportional loading and compatibility with their efficacy and toxicity characteristics are important aspects to consider [316,317]; develop biodegradable carriers of natural origin, including chitosan [318], silk fibroin [319] and gelatin [320], as well as enzyme-responsive polymer systems, which can achieve precise degradation under specific conditions in vivo, thereby reducing the risk of long-term retention.

### 5.2. Multifunctional Integrated Intelligent Nano-Platform

Future nanotherapeutic systems should not be limited to single drug delivery but should develop into an “integrated diagnosis and treatment” platform. For example, multiple functions such as drug release, image tracking, stimulus response, and immune regulation can be integrated into a nanostructure to achieve real-time therapeutic feedback and precise regulation. In recent years, combination modes such as photothermal + chemotherapy and photodynamic + immunotherapy have achieved good results in animal models, proving that multimodal synergistic strategies can effectively overcome tumor heterogeneity and drug resistance [321,322,323]. Future nanosystems are expected to place greater emphasis on responsive designs tailored to the specific characteristics of the TME, including pH gradients, redox conditions, and enzyme expression levels. For example, by constructing reduction-sensitive or low pH degradable carriers, drugs can be specifically released in tumor cells. In addition, new targeting strategies, such as using tumor exosome surface markers for homologous targeting or using specific receptors for recognition, improve targeting efficiency [34].

### 5.3. Green and Sustainable Manufacturing and Scale-Up

At present, the preparation process of nanomaterials generally has problems such as large consumption of organic solvents, high energy consumption, and difficulty in degradation, and it is urgent to transform to green synthesis and sustainable manufacturing. New generation preparation technologies such as microfluidics [324] and the supercritical fluid method [325] can greatly improve the particle size control accuracy and reaction efficiency, reduce batch differences, and have the potential to transform to GMP-level processes. In addition, “green nanotechnology” that uses plant extracts, microbial fermentation, or natural macromolecules to assist in the synthesis of metal nanoparticles is gradually recognized as a new low-toxic and low-cost preparation route [326,327,328].

### 5.4. Big Data and Artificial Intelligence-Assisted Design

Drug release kinetics is one of the most important parameters for characterizing nanomaterials, as it significantly affects potential therapeutic effects, toxicity, and possible off-target effects [329]. Nanosensors can be used to monitor the release and distribution of drugs in patients [330]. Combined with artificial intelligence (AI) analysis, the drug delivery system can be optimized to ensure effective drug concentration in target tissues, reduce toxic side effects, and improve therapeutic effects [331].

Utilizing artificial intelligence and machine learning methods, researchers can develop predictive models that correlate the structural characteristics of nanomaterials with their functional performance using available data. This approach enables the modeling and analysis of various parameters, including drug release profiles, toxicity assessments, and cellular uptake efficiency [332,333,334]. Chi et al. [335] first proposed and verified a rapid pathology diagnosis system based on perovskite nanocrystal probes and machine learning image processing, which is suitable for a variety of cancers, has high sensitivity, speed, and classification accuracy, and is of great value in improving clinical diagnosis efficiency and postoperative patient prognosis. Mi et al. [336] built a machine learning prediction platform based on the Nano Tumor database, especially a DNN model with excellent performance, to achieve quantitative prediction of the delivery efficiency of nanoparticles in tumors and multiple organs (R^2^ ≥ 0.41).

Meanwhile, AI holds great potential in accelerating the development of high-performance nanoplatforms by facilitating material screening, optimizing surface modification strategies, and predicting the structures of imaging agents [337,338]. In many cancer cases, the unpredictable pharmacokinetic behavior of nanoformulated drug delivery systems often leads to drug failure in clinical trials. In order to solve this problem and promote the rational design of nanomedicines, it becomes crucial to predict and consider the main pharmacokinetic parameters when planning the synthesis process and designing materials. In this context, Kaminskas et al. [339] introduced a web-based platform specifically for predicting the pharmacokinetics of dendritic molecule-based therapies. The platform evaluates dendritic structure and uses the cutoff scanning matrix (CSM) method to estimate key parameters, including half-life, clearance, distribution volume, and dose recovered in the liver and urine.

## 6. Conclusions

This review summarizes the diverse applications of nanomaterials in cancer therapy, including drug delivery, photothermal therapy, and immunotherapy. Nanodrug delivery systems improve the pharmacokinetic and pharmacodynamic properties of chemotherapeutics, enhance tumor targeting and accumulation, and reduce off-target toxicity via encapsulation. Additionally, nanomaterial-based photothermal and photodynamic therapies leverage the unique light absorption and conversion properties of nanostructures for minimally invasive, localized tumor ablation, providing novel strategies.

Despite their promising potential, nanomaterials in cancer therapy face key challenges in safety and clinical translation. While nanotechnology may revolutionize cancer treatment, technical and clinical barriers hinder its widespread use. Though nanomaterials offer novel strategies to address longstanding oncology challenges, unresolved issues persist—particularly in biosafety. Despite studies on nanomaterial-biological system interactions, their in vivo long-term fate, metabolic pathways, toxicological effects, and impacts on human health and the environment remain incompletely understood [285]. While existing reviews have addressed the acute toxicity of nanomaterials, analyses of the structure-activity relationship concerning “size-surface modification-long-term accumulation toxicity” remain fragmented [340]. This review integrates toxicological data from diverse materials—metallic, organic, and carbon-based—to propose a “phased safety assessment framework” that offers greater practical applicability than previous approaches.

Owing to their small size, nanomaterials can penetrate biological barriers to enter cells, tissues, and organs. However, some may accumulate in the body, potentially inducing inflammatory responses, immunotoxicity, or genotoxicity [286]. For instance, nanoparticles may activate the immune system, causing excessive cytokine release and systemic inflammatory response syndrome, or interact with intracellular genetic material to induce genotoxic effects that disrupt normal cellular function [341]. These biosafety concerns remain a major obstacle to the large-scale clinical application of nanomaterials.

Additionally, nanomaterial synthesis requires further optimization: current fabrication methods rarely achieve highly uniform, stable products. While particle size, shape, and surface characteristics are critical to therapeutic performance, their precise control remains challenging in practice, causing significant batch-to-batch variability and inconsistent therapeutic outcomes [342]. Although diverse surface modifications (e.g., polymer coatings, ligand conjugation) have been developed to enhance biocompatibility, targeting, and circulation time, challenges remain in designing nanocarriers that selectively recognize and efficiently internalize into tumor cells while evading premature immune clearance [343,344].

In clinical translation, the shift from laboratory research to clinical application faces numerous challenges. Physiological and pathological differences between animal models and humans hinder the direct translation of promising preclinical results to clinical efficacy [343]. Animal models poorly replicate the complexity of the human physiological environment, including differences in immune systems, metabolic pathways, and tumor microenvironment heterogeneity. Additionally, nanomaterial regulatory frameworks and standards remain underdeveloped, with a lack of unified quality control, evaluation systems, and consistent guidelines for quality standards, safety assessment, and clinical protocols—posing significant barriers to clinical approval and regulatory oversight [345,346]. Some literature focuses on macro-level policy analysis, while this review combines specific case studies, such as the application of PDX models and the integration of multi-center clinical data, to propose a “foundational-clinical pathway”, while supplementing practical recommendations for “collaborative standard-setting between industry and regulators” [347]. This approach offers greater implementation feasibility than previous studies and fills a gap in existing reviews regarding “translational implementation strategies.”

To advance nanomaterial applications in cancer therapy, future research should focus on key directions. First, a comprehensive evaluation of nanomaterial biosafety is essential, including developing robust in vitro/in vivo models and long-term monitoring systems [348]. Second, continuous optimization of nanomaterial synthesis techniques is needed for precise, controllable production. Advanced approaches (e.g., self-assembly, template methods) should be used to accurately regulate nanomaterial size, shape, and surface characteristics, improving batch-to-batch consistency and quality control [276]. In parallel, the integration of computer-aided design and high-throughput screening could significantly improve the design of surface modification strategies, enhancing tumor-targeting efficiency. Strengthening the link between preclinical and clinical research is also critical. The establishment of patient-derived xenograft (PDX) models that more closely mimic human tumor biology and microenvironments can help bridge the gap between basic research and clinical application [349]. Additionally, it is crucial to promote the development and refinement of relevant regulatory policies and standards. Multidisciplinary collaboration among experts in nanotechnology, oncology, pharmacology, and regulatory science is needed to develop guidelines tailored to the unique properties of nanomaterials and their clinical requirements.

Although nanomaterials have shown significant progress in cancer therapy, their application remains at an early developmental stage. Addressing the existing challenges will require close collaboration across multiple disciplines, including materials science, biology, medicine, and pharmacy [32]. A deeper exploration of the interface between nanomaterials and tumor biology is essential. Elucidating the mechanisms of interaction between nanomaterials and tumor cells at the molecular, cellular, and tissue levels will help to fully realize their potential and deliver more effective therapeutic strategies for cancer patients, ultimately achieving broader clinical translation [350].

## Figures and Tables

**Figure 1 biomedicines-13-02666-f001:**
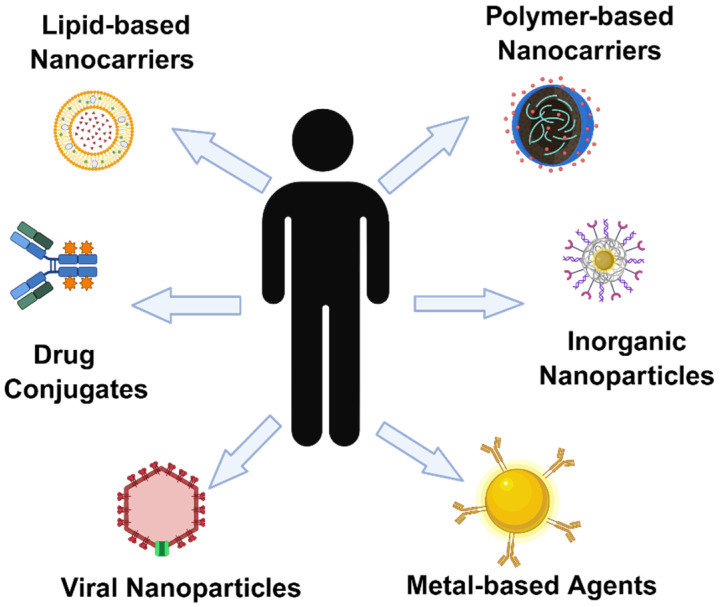
Schematic illustration of established nanotherapeutic platforms. Different nanomedicine products, such as drug conjugates, lipid-based nanocarriers, polymer-based nanocarriers, inorganic nanoparticles, metal-based agents, and viral nanoparticles, are used in clinical cancer care.

**Figure 2 biomedicines-13-02666-f002:**
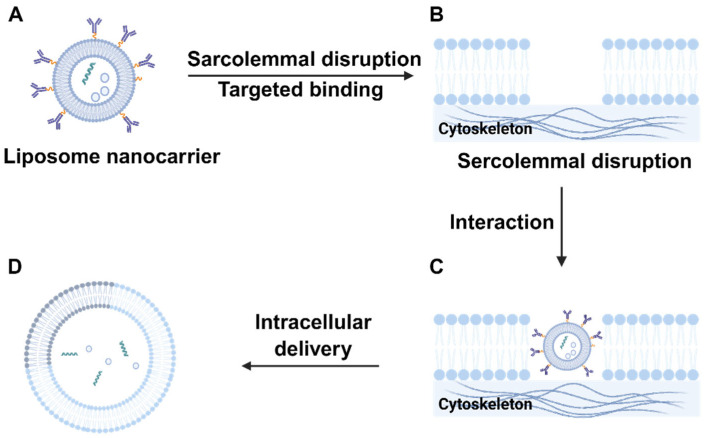
Visualization of intracellular delivery stages for bioactive compounds via the carriers, elucidating their role in mitigating drug toxicity. (**A**) Liposome nanocarrier; (**B**) Sarcolemmal disruption initiated to enable carrier-mediated delivery; (**C**) Interaction dynamics between carriers, bioactive compounds, and cellular machinery; (**D**) Intracellular delivery completion, underscoring reduced toxic effects.

**Figure 3 biomedicines-13-02666-f003:**
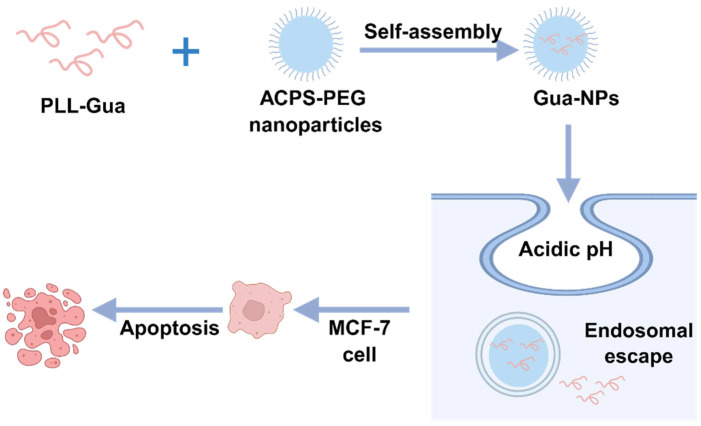
The PEG as a vehicle for ACP delivery. Schematic representation of Gua-NPs formation and their anticancer mechanism. PLL-Gua and PLL-PEG nanoparticles self-assemble into Gua-NPs, which enter cancer cells via endocytosis. In the acidic endosomal environment, Gua is released, leading to endosomal escape and apoptosis induction in cancer cells.

**Figure 4 biomedicines-13-02666-f004:**
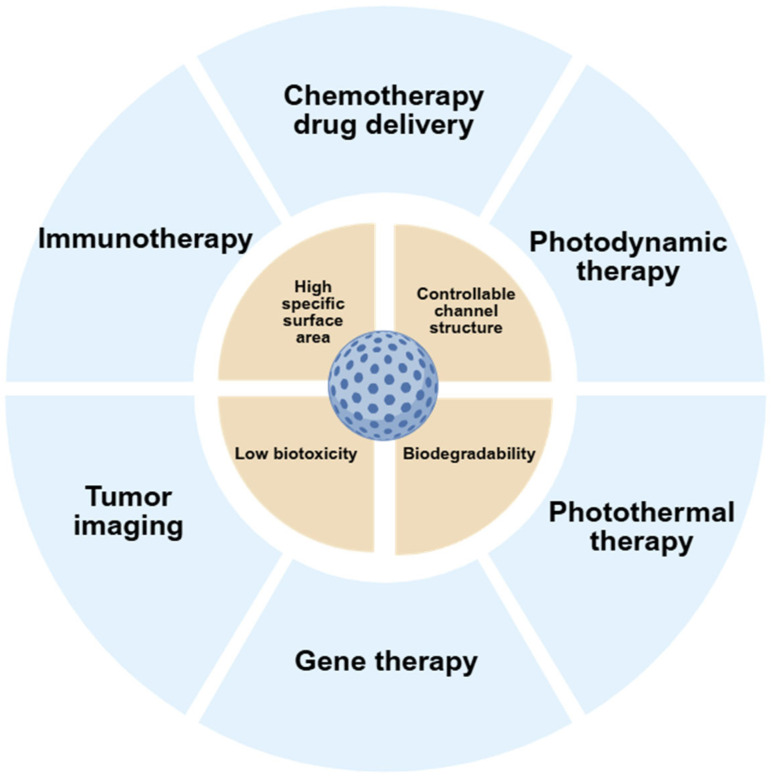
Schematic illustration of the biomedical applications and intrinsic structural advantages of nanosilica. The central region highlights key physicochemical properties of nanosilica, including high specific surface area, tunable aperture structure, surface-modified low-toxicity substances, and intrinsic biodegradability. These features collectively endow nanosilica with exceptional versatility in a range of biomedical applications. The outer segments demonstrate their utility in drug delivery, gene delivery, tumor imaging, bioimaging, photodynamic therapy, photothermal therapy, and immunotherapy.

**Figure 5 biomedicines-13-02666-f005:**
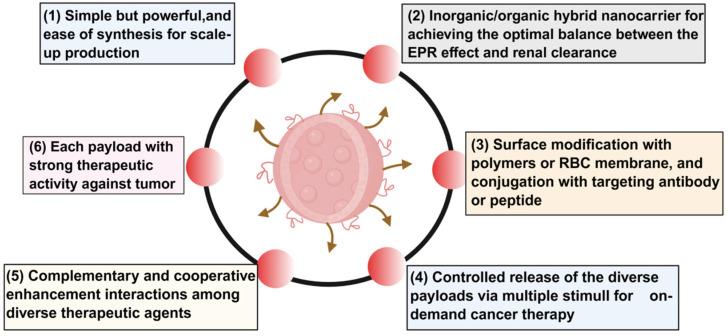
Schematic representation of the key design considerations for multifunctional nanocarriers engineered to achieve synergistic therapeutic outcomes through the integration of multiple treatment modalities. The design strategy encompasses: (1) facile and scalable synthesis to ensure practical applicability; (2) construction of inorganic/organic hybrid architectures to optimize the balance between enhanced permeability and retention (EPR) effects and renal clearance; (3) surface modification with polymers or red blood cell (RBC) membranes, along with conjugation of targeting ligands (e.g., antibodies or peptides) to improve tumor selectivity; (4) implementation of controlled and stimuli-responsive drug release mechanisms to enable site-specific and temporally regulated therapy; (5) promotion of cooperative interactions among distinct therapeutic modalities to enhance synergistic efficacy. (6) loading of diverse therapeutic agents with potent anticancer efficacy. Collectively, these principles guide the rational design of nanotherapeutic platforms with high translational potential in cancer treatment.

**Table 1 biomedicines-13-02666-t001:** Common types of nanocarriers used in tumor therapy, their advantages and characteristics.

Types of Nanocarriers	Core Components/ Structure	Key Advantages	Key Features	Typical Application Scenarios	Reference
Metal Nanocarriers	Gold, silver, iron oxide, titanium dioxide, and other metals/metal oxides; predominantly spherical, cage-like, or core–shell structures.	Excellent photothermal/photodynamic properties; Easy surface functionalization; Controllable biocompatibility.	Adjustable size; Combines therapeutic and imaging functions; High photothermal conversion efficiency.	Photothermal therapy for solid tumors; Combination therapy; Tumor imaging-guided therapy.	[159,160,161]
Polymeric nanocarriers	Synthetic polymers or natural polymers, typically in the form of micelles, microspheres, or nanoparticles.	High drug encapsulation efficiency; Controllable degradation rate; Strong long-term cycling capability	Surface charge is tunable. Enables stimuli-responsive drug release; High biocompatibility.	Extended-release delivery of chemotherapy drugs; Gene therapy; Oral drug solubilization.	[162,163,164,165,166]
Lipid-based nanocarriers	Phospholipids, cholesterol, ionizable lipids; primarily liposomes, lipid nanoparticles, micelles	Excellent biocompatibility; Capable of encapsulating both hydrophilic and hydrophobic drugs; Suitable for large-scale production.	Uniform size, with extended circulation half-life achievable through PEG modification; LNP enables efficient delivery of nucleic acid therapeutics; Flexible adjustment of lipid composition.	Targeted delivery of chemotherapy drugs; Gene therapy; Immunotherapy	[167,168,169,170]
Carbon-based nanocarriers	Graphene, carbon nanotubes, mesoporous carbon, fullerenes; predominantly sheet-like, tubular, or porous structures.	Large specific surface area with high drug loading capacity; Excellent electronic conductivity suitable for photothermal/PDT; Easily functionalized surface.	High physical stability; Mesoporous carbon possesses controllable pore sizes, facilitating controlled drug release; Graphene quantum dots exhibit fluorescent properties, enabling their use in fluorescence imaging.	High-dose chemotherapy drug delivery; Photothermal therapy; Multimodal tumor imaging.	[171,172,173,174]

**Table 2 biomedicines-13-02666-t002:** Nano-Material-Mediated Multimodal Combined Tumor Therapy Approaches and Characteristics.

Combination Therapy	Core Principle	Selection of Nanomaterial Carriers	Advantages	Limitations	Reference
Chemotherapy + Photothermal Therapy	Nanocarriers encapsulate chemotherapy drugs; the carriers themselves or loaded with photothermal agents generate heat upon near-infrared irradiation to directly destroy tumor cells; the thermal effect enhances cell membrane permeability, promotes endocytosis of chemotherapy drugs, and reverses drug resistance.	Gold nanocages, PLGA-PEG nanoparticles, graphene quantum dots	Synergistically enhances lethal effects; Light-controlled delivery enables precise spatiotemporal release, minimizing exposure of normal tissues.	The penetration depth of photothermal therapy is limited, rendering it ineffective for deep-seated tumors; high temperatures (>45 °C) may damage surrounding healthy tissues; prolonged photothermal exposure may induce thermal tolerance in tumor cells.	[261,262,263]
Chemotherapy + PDT	Nanocarriers co-load chemotherapy drugs and photosensitizers; light of specific wavelengths excites the photosensitizer to generate singlet oxygen, oxidatively damaging tumor cells; chemotherapy drugs induce apoptosis, synergizing with the oxidative stress induced by PDT.	Liposomes, mesoporous silica nanoparticles	Dual-mechanism killing reduces drug resistance risk; photosensitizer targets tumor enrichment to minimize systemic phototoxicity; fluorescence imaging enables real-time monitoring of photosensitizer distribution.	Photosensitizers require an oxygen-rich environment for activation, limiting their efficacy in hypoxic tumors; insufficient light penetration depth necessitates fiber-optic intervention; strict photoprotection is required to prevent skin phototoxicity.	[264,265,266]
Chemotherapy + Immunotherapy	Nanocarriers encapsulate chemotherapy drugs and immunomodulators; chemotherapy induces immunogenic cell death, releasing tumor-associated antigens; immunomodulators activate dendritic cells and cytotoxic T cells, suppressing immune escape.	Polyethylene glycol–polycaprolactone micelles, dendritic polymerization	Chemotherapy transforms “cold tumors” into “hot tumors,” enhancing immunotherapy response; co-delivery reduces immunosuppressant dosage and alleviates systemic toxicity; long-term immune memory effect lowers recurrence rates.	Immune-related adverse reactions may still occur; Tumor microenvironment immunosuppression may diminish efficacy; Individual immune heterogeneity leads to significant response variability.	[267,268,269,270]
PDT + Immunotherapy	Nano-photothermal agents generate heat upon laser irradiation, directly destroying tumors while releasing tumor-associated antigens TAAs. Heat shock proteins assist antigen presentation, and when combined with immune checkpoint inhibitors or CAR-T cells, they activate systemic antitumor immunity.	Gold Nanoparticles, Black Phosphorus Quantum Dots	Photothermal therapy exhibits no drug resistance and synergizes with the immune system to achieve “local ablation + systemic tumor control”; the thermal effect promotes immune cell infiltration; it avoids the damage chemotherapy inflicts on immune cells.	The photothermal range is limited, requiring precise tumor localization; high temperatures may disrupt antigen structures, weakening immunogenicity; photothermal therapy alone struggles to activate effective immunity against immune-cold tumors.	[261,271,272,273]
Photothermal + Photodynamic + Chemotherapy Triple Therapy	The nanocarrier integrates photothermal agents, photosensitizers, and chemotherapeutic drugs; near-infrared light excites photothermal heating and singlet oxygen production while simultaneously releasing chemotherapeutic drugs. This triple-mechanism synergy targets both primary tumors and micrometastases.	Mesoporous carbon nanospheres	Multiple mechanisms work together to minimize the risk of drug resistance; A single light source triggers multiple treatments for simplified operation; Suitable for refractory tumors.	The carrier synthesis is complex, with poor batch-to-batch consistency; multiple therapeutic units may interfere with each other; adverse effects may accumulate, necessitating strict dose control.	[274,275]

## Data Availability

No new data were created or analyzed in this study. Data sharing is not applicable to this article.

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
