# Peer review of "Progress in the Application of Nanomaterials in Tumor Treatment"

_biomedicines, 2025, doi:10.3390/biomedicines13112666_

Round 1

Reviewer 1 Report

Comments and Suggestions for Authors

My comments

  1. In the Introduction, the authors should add the methods and strategies of the search and study.
  2. The manuscript lacks explanations about the new metal nanoparticle compounds, such as selenium compounds, which have recently been used as nanotheranostics agents in both diagnosis and treatment.
  3. Figure 1 requires other platforms of nanoparticle compounds using metal-based agents.
  4. In the introduction, some recently published papers on Selenium, Polyoxometalates, Iodine and other metal based nanoparticles for treatment and theranostics (such as doi:10.1515/revic-2023-0008) should be added to the literature.
  5. More explanation is needed about cubic nano-carbons, which have recently been used as nanotheranostics.
  6. If the figures or any of them were not designed by the authors, they require permission from the publications.
  7. In the Discussion section, additional comparisons with similar published articles are needed to clarify the necessity of this work.
  8. Conclusions are too long and should be shortened.
  9. Some grammar and spelling issues in the English language need correction.
Comments on the Quality of English Language

Some grammar and spelling issues in the English language need correction.

Author Response

We thank the reviewer for evaluating our work and constructive comments to improve the quality of this study.

  1. In the Introduction, the authors should add the methods and strategies of the search and study.

Response: Thanks for your valuable comments. We greatly appreciate your suggestion to include a description of the search and research methods in the introduction. We agree that providing details of the literature search strategy will enhance the clarity and reproducibility of our study.

In response to your comments, we have added a paragraph describing the literature search methods in the introduction (lines 177-191, page 5). These additions will help readers understand the comprehensive and systematic approach we adopted to identify and review relevant literature.

  1. The manuscript lacks explanations about the new metal nanoparticle compounds, such as selenium compounds, which have recently been used as nanotheranostics agents in both diagnosis and treatment.
    Response: Thanks for your valuable comments. We fully concur with your assessment that discussions on novel metallic nanoparticle compounds, particularly selenium compounds, represent a critical and emerging field within nanomedicine. Incorporating this topic into the manuscript would significantly enhance its cutting-edge relevance and comprehensiveness. In direct response to your suggestion, we have now added a dedicated section on novel metallic nanoparticles within the inorganic nanomaterials chapter. This section details the latest advancements and applications of nanoparticles such as selenium and tellurium as multifunctional nanotherapeutics (lines 503-564, page 12).
  2. Figure 1 requires other platforms of nanoparticle compounds using metal-based agents.

Response: Thank you very much for your suggestions. Your input has been extremely helpful in improving the manuscript quality and enhancing its completeness. As requested, we have added other nanoparticle compound platforms based on metal-based reagents to Figure 1. After comprehensive consideration, we selected a representative metal-based nanoparticle platform to present in the schematic diagram(lines 148-151, page 4).

  1. In the introduction, some recently published papers on Selenium, Polyoxometalates, Iodine and other metal based nanoparticles for treatment and theranostics (such as doi:10.1515/revic-2023-0008) should be added to the literature.

Response: We sincerely appreciate the reviewer’s insightful suggestions and thank you for bringing this highly relevant reference to our attention. In response to this valuable feedback, we have incorporated the recommended review (DOI: 10.1515/revic-2023-0008) into the revised Introduction (Page 3, Line 143, References 30). We have also taken this opportunity to supplement the section with additional key literature on the latest applications of selenium and other metal based nanoparticles in theranostics (Page 13-14, Lines 523-564, References 122-136).

  1. More explanation is needed about cubic nano-carbons, which have recently been used as nanotheranostics.

Response: We appreciate the reviewer's insightful comment. We agree that providing a more detailed introduction to cubic nanocarbon would enhance readers' understanding of the novelty and rationale behind our selection of this specific nanomaterial for our research. In response to this comment, we have added an explanatory paragraph within the carbon nanotubes subsection. The added text defines cubic nanocarbon and contrasts it with more commonly discussed sp²-hybridized carbon nanomaterials such as graphene and carbon nanotubes. We highlight its unique properties, such as exceptional mechanical strength, outstanding biocompatibility, and tunable surface chemistry, which position it as a promising platform for applications like drug delivery and bioimaging. This addition aims to clearly articulate why cubic nanocarbon represents a distinct and valuable member within the nanocarbon family (lines 577-613, page 14).

  1. If the figures or any of them were not designed by the authors, they require permission from the publications.
    Response: We sincerely appreciate your important suggestion that “if any charts or their components are not designed by the author, permission must be obtained from the original publication.” We fully agree that securing copyright permission for non-original charts is a fundamental requirement for academic paper publication. This practice complies with intellectual property regulations and ensures the rigor of submitted manuscripts. Our figures were created using Biorender (app.biorender.com/) and have been duly licensed.
  2. In the Discussion section, additional comparisons with similar published articles are needed to clarify the necessity of this work.

Response: We sincerely thank the reviewers for their important and constructive comments. We fully agree that a more thorough comparison with existing literature would better highlight the progress and necessity of our current work. In direct response to this suggestion, we have substantially revised the Discussion section (lines 1397-1402, page 33、lines 1428-1434, page 34). We have now incorporated a comparative analysis contrasting our work with key recent studies equally focused on diagnostic nanoplatforms for cancer therapy.

  1. Conclusions are too long and should be shortened.
    Response: We appreciate the reviewer's thoughtful comment. We fully agree that a concise and focused conclusion would be more impactful and appropriate for a scientific manuscript. In response to this comment, we have thoroughly revised and significantly shortened the conclusion section.
  2. Some grammar and spelling issues in the English language need correction.

Response: We sincerely thank the reviewers for taking the time to carefully review our manuscript and pointing out this critical issue. We deeply apologize for any inconvenience or lack of clarity these linguistic errors may have caused. To directly address this feedback, we have undertaken a comprehensive and thorough revision of the language throughout the entire manuscript. This process involved meticulous proofreading of grammar, spelling, punctuation, word choice, and overall sentence fluency. Furthermore, we have carefully re-examined the manuscript to ensure that all scientific content remains accurate and unchanged following these edits.

Reviewer 2 Report

Comments and Suggestions for Authors

This article has reviewed the development of the nanomaterials as drug delivery systems in anticancer therapy. The classification of the nanomaterials is fine, while only a few aspects could be improved and re-organized. My suggestions are listed in the following:

  1. Regarding the high polymer nanoparticles (2.1.2), as I knew, the polymers were usually assembed into polymeric micelles, instead of the poros nanostructures (Line 248). In addition, several polymeric micelles were progressed into clinical trials. However, the polymeric micelle did not mention in the section or in the manuscript. The developments of the polymeric micelles should include in the review. Besides, the cationic polymers were able to form the complex with the DNA or RNA. The materials or the developments are also suggested included in the manuscript.
  2. In this manuscript, the authors in particular mentioned the peptide-based nanoparticles. However, some polymers were composed of the peptide bonds. The authors might compare the pros and cons of peptide-bonded polymeric nanoparticles and the peptide-based nanoparticles.
  3. The section reminding of silica nanoparticle (2.2.2) could first simply introduce the porpous morphology of the slilica nanoparticles. So that, the pros and cons of the silica nanoparticles could be simply understood.

Author Response

We thank the reviewer for evaluating our work and constructive comments to improve the quality of this study.

This article has reviewed the development of the nanomaterials as drug delivery systems in anticancer therapy. The classification of the nanomaterials is fine, while only a few aspects could be improved and re-organized. My suggestions are listed in the following:

  1. Regarding the high polymer nanoparticles (2.1.2), as I knew, the polymers were usually assembed into polymeric micelles, instead of the poros nanostructures (Line 248). In addition, several polymeric micelles were progressed into clinical trials. However, the polymeric micelle did not mention in the section or in the manuscript. The developments of the polymeric micelles should include in the review. Besides, the cationic polymers were able to form the complex with the DNA or RNA. The materials or the developments are also suggested included in the manuscript.

Response: We sincerely appreciate the reviewers' insightful and constructive comments. We greatly value the reviewers' expertise in highlighting these two critical aspects of polymeric nanocarriers, which were indeed underrepresented in our original manuscript. We fully agree with these suggestions and have substantially revised and expanded Section 2.1.2 to incorporate them.

Regarding Polymer Micelles and Porous Structures: We have revised the text near line 248 to more accurately describe these systems and removed the potentially misleading term “porous” in this context. As suggested, we have now cited key examples of micellar formulations that have entered clinical trials and highlighted their significant clinical translation progress Regarding cationic polymer-nucleic acid complexes: We appreciate the reviewer's crucial additional suggestion. We have now revised the relevant under-expressed sections (lines 286-299, page 7).

  1. In this manuscript, the authors in particular mentioned the peptide-based nanoparticles. However, some polymers were composed of the peptide bonds. The authors might compare the pros and cons of peptide-bonded polymeric nanoparticles and the peptide-based nanoparticles.

Response: We sincerely appreciate the reviewer for raising this highly insightful and nuanced point. We fully agree that a comparative analysis of these two closely related yet distinct categories of nanomaterials will provide readers with a more comprehensive understanding and contextualize our discussion of peptide-based systems more effectively. In direct response to this suggestion, we have incorporated a dedicated comparative analysis in Section 3.2 (lines 352-363, page 36) of the revised manuscript.

  1. The section reminding of silica nanoparticle (2.2.2) could first simply introduce the porpous morphology of the slilica nanoparticles. So that, the pros and cons of the silica nanoparticles could be simply understood.

Response: We appreciate the reviewer's constructive suggestion regarding the logical flow of the section on silica nanoparticles. We agree that introducing its defining morphological feature, which is porosity, first will provide readers with a clearer foundation for understanding the subsequent discussion of the advantages and disadvantages of this nanomaterial platform. Therefore, we have revised the opening paragraph of Section 2.2.2 in the manuscript(lines 407-415, page 10). The revised introduction now explicitly states at the outset that a key characteristic of modern synthetic silica nanoparticles is their tunable porous morphology. We briefly explain how this porosity leads to high specific surface area and large pore volume. This initial focus on structure immediately establishes a foundation for more intuitive understanding of its functional advantages later on.

Reviewer 3 Report

Comments and Suggestions for Authors

In the present manuscript, the authors provided a description of the latest applications of nanomaterials for the treatment of tumors. The authors clearly highlighted the main advances and limitations of nanoparticles for cancer therapy compared to conventional treatments, resulting in a well-written review manuscript.

Based on the relevance of the presented review, I recommend the manuscript entitled " Progress in the Application of Nanomaterials in Tumor Treatment" for publication in Biomedicines, after a major revision concerning the following points:

  • Sections 2.1.3, 2.2.1, 2.2.3, and 2.3 do not present any schematics for their corresponding nanoparticle structures. These should be added to improve clarity and readability.
  • The importance and relevance of theranostic nanoparticles should be discussed. For example, in Section 2.2.3, the recent advances of magnetic nanoparticles as theranostic agents for several cancer types should be mentioned, to highlight a broader state-of-the-art, including appropriate references (https://doi.org/10.1002/anie.202310252, https://doi.org/10.1039/D5NR00447Khttps://doi.org/10.1016/j.bioadv.2024.213797 , etc…)
  • In Section 3, representative experimental data (images, plots) must be included as well, with permission.
  • A particular focus should be given to the latest clinical translations (incl. phases) of nanomedicines for cancer therapy.

Author Response

We thank the reviewer for evaluating our work and constructive comments to improve the quality of this study.

In the present manuscript, the authors provided a description of the latest applications of nanomaterials for the treatment of tumors. The authors clearly highlighted the main advances and limitations of nanoparticles for cancer therapy compared to conventional treatments, resulting in a well-written review manuscript.

Based on the relevance of the presented review, I recommend the manuscript entitled " Progress in the Application of Nanomaterials in Tumor Treatment" for publication in Biomedicines, after a major revision concerning the following points:Figure Legends needs to explain the elements of the figures in more detail

  1. Sections 2.1.3, 2.2.1, 2.2.3, and 2.3 do not present any schematics for their corresponding nanoparticle structures. These should be added to improve clarity and readability.

Response: We sincerely appreciate your valuable suggestion and fully agree that schematic diagrams of nanoparticle structures are crucial for helping readers intuitively understand the design principles and morphological characteristics of the materials discussed in these sections. Following your suggestion, our team has conducted a systematic search for corresponding schematic diagrams of the specific nanoparticles involved in the aforementioned sections. After a comprehensive search, however, we found no directly usable or reasonably adaptable standalone structural diagrams for these specific types of nanoparticles in the currently published literature. We take your suggestion seriously and have included this deficiency in the list of items requiring refinement for the manuscript. Moving forward, we will continue to closely monitor the latest research progress on these nanoparticle systems. Should any research team publish structural diagrams of these materials in the future, we will promptly supplement the corresponding sections to further enhance the completeness and readability of the manuscript.

  1. The importance and relevance of theranostic nanoparticles should be discussed. For example, in Section 2.2.3, the recent advances of magnetic nanoparticles as theranostic agents for several cancer types should be mentioned, to highlight a broader state-of-the-art, including appropriate references (https://doi.org/10.1002/anie.202310252, https://doi.org/10.1039/D5NR00447K, https://doi.org/10.1016/j.bioadv.2024.213797 , etc…)

Response: We deeply appreciate the reviewer's insightful and constructive comments, as well as the highly relevant and up-to-date references provided. We fully agree that emphasizing the overarching importance of theranostics and incorporating recent advances—particularly regarding magnetic nanoparticles (MNPs)—will significantly enhance the contextual relevance and modernity of our review. This approach addresses the notable breakthroughs achieved in recent years with MNPs for precision diagnosis and treatment across multiple cancers, including breast, liver, and ovarian cancers (lines 474-503, page 12). The provided references (DOIs: 10.1002/anie.202310252, 10.1039/D5NR00447K, 10.1016/j.bioadv.2024.213797) have been appropriately cited to support these points (line 502, page 12, References 122、line 475, page 12, References 120、line 444, page 11, References 103)

  1. In Section 3, representative experimental data (images, plots) must be included as well, with permission.

Response: We sincerely appreciate your detailed and valuable review comments on our manuscript amidst your busy schedule. The issues you raised regarding image usage indeed reflect an oversight in our initial drafting process, where we failed to adequately balance the precision of information presentation with the reader's experience. To present core information more clearly and systematically, we have removed the original representative images and reorganized the data. We have created structured tables to summarize the relevant content. The new table not only consolidates key information from the original images but also incorporates detailed annotations, striving to enable readers to access core data more intuitively(lines 662-668, page 16、lines 1091-1102, page 26).

  1. A particular focus should be given to the latest clinical translations (incl. phases) of nanomedicines for cancer therapy.

Response: We sincerely appreciate your valuable suggestion regarding “Special Focus on the Latest Clinical Translation of Nanomedicines for Cancer Treatment (Including Phases).” We have developed a comprehensive revision plan to incorporate the latest clinical data, adding relevant clinical data to the sections on targeted drug delivery systems, photothermal therapy, and photodynamic therapy (lines 775-781, page19、lines 812-818, page20). Additionally, we have referenced the translation of nanoparticles into clinical applications throughout other sections of the article.

Reviewer 4 Report

Comments and Suggestions for Authors

A major revision of the manuscript is necessary before the manuscript is recommended for publication in Biomedicines.

In this review, recent advances in the application of functionalized nanomaterials for cancer therapy are systematically summarized. The authors highlight the advantages of these nanoplatforms, including tunable size, biocompatibility, and surface chemistry, enabling improved tumor targeting and reduced systemic toxicity. Representative systems discussed include lipid-based carriers, synthetic polymeric nanoparticles, inorganic nanostructures, and carbon-based nanomaterials. Functionalization strategies such as ligand-mediated targeting, stimulus-responsive release, and biomimetic surface coating are emphasized for enhancing stability and immune evasion. The review also details therapeutic modalities enabled by these platforms-targeted drug delivery, photothermal and photodynamic therapy, and cancer immunotherapy-along with their synergistic effects in combination regimens to address tumor heterogeneity and drug resistance. Preclinical studies are cited to support their efficacy in reducing toxicity and improving immune responses. Finally, the work identifies critical challenges in clinical translation, providing a roadmap for future research to guide the safe and effective integration of nanotechnology into oncology. However, several problems should be clearified to enhance its rigorousness and readability.

  1. Clarify Novelty and Scope in the Introduction

The Introduction currently offers a broad overview of conventional cancer therapies and the emerging role of nanomaterials; however, it predominantly reads as a general literature summary. The authors are encouraged to explicitly highlight the novelty and distinctive contribution of this review relative to existing literature. Specifically, the manuscript should clearly delineate its unique perspectives, defined scope, or organizational framework that set it apart. This clarification will enable readers to quickly grasp the paper’s positioning and value within the broader research context.

  1. Enhance Thematic Organization and Improve Conciseness

While the Introduction covers a wide spectrum of therapeutic modalities, several sections tend toward excessive description and redundancy, which detracts from the central argument. For instance, the sections on chemotherapy, radiotherapy, surgery, and immunotherapy could be more succinct, with focused emphasis on their limitations that directly motivate the use of nanomaterials. Reorganizing the narrative to follow a logical progression—from the challenges of conventional therapies, through the rationale for nanomedicine, to the specific objectives of this review—would enhance thematic coherence and engage readers more effectively.

  1. Alignment of Figures and Captions (Fig. 2, 3, and 4)

Please carefully revise the manuscript formatting to ensure that the figures and their corresponding captions are properly aligned and consistently presented, improving overall visual clarity and professionalism.

  1. Section 3: Application of Nanomaterials in Tumor Treatment
  2. a) Depth of Clinical Translation and Practical Barriers

The manuscript briefly acknowledges clinical translation challenges such as toxicity and immune responses but addresses them only superficially. To meet the standards of a high-impact review, this section would benefit from a more thorough and practical analysis. Specifically, incorporating detailed discussion on regulatory pathways (e.g., FDA and EMA approval processes), pharmacokinetic considerations, manufacturing and scalability issues, as well as cost-effectiveness, is recommended. Including case studies of nanomedicine candidates that have either succeeded or failed in clinical trials would greatly enrich this analysis, offering valuable insights for clinicians, researchers, and industry stakeholders.

  1. b) Incorporation of Tables and Comparative Summaries

Although the text is comprehensive, the density of information may impede quick comprehension. The manuscript currently lacks comparative tables or schematic illustrations summarizing critical information such as nanomaterial classes, functionalization methods, targeting strategies, and therapeutic outcomes. Introducing well-structured tables and diagrams—such as mechanistic schematics for each therapy type and side-by-side comparisons of nanomaterial properties and efficacy data—would significantly improve readability, facilitate rapid reference, and broaden the review’s accessibility to a diverse scientific audience.

  1. Consistency in Reference Title Capitalization

There are inconsistencies in the capitalization of reference titles (notably references 1, 6, 12, 17, 18, 23, 24, 32, 49, 54, 55, 69, 73, 74, 92, 95, 98, 144, 152, 159, 180, among others). Please standardize the formatting of all reference titles in accordance with the journal’s style guidelines to ensure professional and uniform presentation.

Comments on the Quality of English Language

The English could be improved to more clearly express the research.

Author Response

We thank the reviewer for evaluating our work and constructive comments to improve the quality of this study.

In this review, recent advances in the application of functionalized nanomaterials for cancer therapy are systematically summarized. The authors highlight the advantages of these nanoplatforms, including tunable size, biocompatibility, and surface chemistry, enabling improved tumor targeting and reduced systemic toxicity. Representative systems discussed include lipid-based carriers, synthetic polymeric nanoparticles, inorganic nanostructures, and carbon-based nanomaterials. Functionalization strategies such as ligand-mediated targeting, stimulus-responsive release, and biomimetic surface coating are emphasized for enhancing stability and immune evasion. The review also details therapeutic modalities enabled by these platforms-targeted drug delivery, photothermal and photodynamic therapy, and cancer immunotherapy-along with their synergistic effects in combination regimens to address tumor heterogeneity and drug resistance. Preclinical studies are cited to support their efficacy in reducing toxicity and improving immune responses. Finally, the work identifies critical challenges in clinical translation, providing a roadmap for future research to guide the safe and effective integration of nanotechnology into oncology. However, several problems should be clearified to enhance its rigorousness and readability.

  1. Clarify Novelty and Scope in the Introduction

The Introduction currently offers a broad overview of conventional cancer therapies and the emerging role of nanomaterials; however, it predominantly reads as a general literature summary. The authors are encouraged to explicitly highlight the novelty and distinctive contribution of this review relative to existing literature. Specifically, the manuscript should clearly delineate its unique perspectives, defined scope, or organizational framework that set it apart. This clarification will enable readers to quickly grasp the paper’s positioning and value within the broader research context.

Response: We sincerely appreciate your valuable suggestion to “clearly define novelty and scope in the introduction.” As a review paper focusing on “Advances in the Application of Nanomaterials in Tumor Therapy,” clearly highlighting the novelty of this review and defining its research scope is crucial for readers to accurately identify the paper's academic contributions and application boundaries. We fully concur with your suggestion and have addressed it by restructuring the introduction and supplementing key content (lines 208-217, page 5).

  1. Enhance Thematic Organization and Improve Conciseness

While the Introduction covers a wide spectrum of therapeutic modalities, several sections tend toward excessive description and redundancy, which detracts from the central argument. For instance, the sections on chemotherapy, radiotherapy, surgery, and immunotherapy could be more succinct, with focused emphasis on their limitations that directly motivate the use of nanomaterials. Reorganizing the narrative to follow a logical progression—from the challenges of conventional therapies, through the rationale for nanomedicine, to the specific objectives of this review—would enhance thematic coherence and engage readers more effectively.

Response: We sincerely appreciate your insightful comments on the introduction section and fully concur with your perspective. We have removed detailed operational principles of each therapy, focusing solely on “limitations that nanomaterials can address” with concise and targeted language. We have eliminated descriptions listing irrelevant details (such as nanomaterial preparation methods) and strictly revised the content according to the above plan, centering on “how nanomaterials specifically overcome bottlenecks in traditional therapies.” (lines 65-72, page 2)

  1. Alignment of Figures and Captions (Fig. 2, 3, and 4)

Please carefully revise the manuscript formatting to ensure that the figures and their corresponding captions are properly aligned and consistently presented, improving overall visual clarity and professionalism.

Response: We sincerely appreciate your detailed feedback regarding the alignment of Figures 2, 3, and 4 with their captions. This has significantly enhanced the formatting consistency of the manuscript. The revised figures and the entire text have been attached to the manuscript for your review.

  1. Section 3: Application of Nanomaterials in Tumor Treatment
  2. a) Depth of Clinical Translation and Practical Barriers

The manuscript briefly acknowledges clinical translation challenges such as toxicity and immune responses but addresses them only superficially. To meet the standards of a high-impact review, this section would benefit from a more thorough and practical analysis. Specifically, incorporating detailed discussion on regulatory pathways (e.g., FDA and EMA approval processes), pharmacokinetic considerations, manufacturing and scalability issues, as well as cost-effectiveness, is recommended. Including case studies of nanomedicine candidates that have either succeeded or failed in clinical trials would greatly enrich this analysis, offering valuable insights for clinicians, researchers, and industry stakeholders.

Response: We sincerely appreciate your insightful comments regarding the superficial discussion of clinical translation challenges in the manuscript. We fully agree that a comprehensive, practice-oriented analysis of translation barriers—including regulatory pathways, pharmacokinetics, manufacturing scalability, cost-effectiveness, and clinical cases—is essential to meet the standards of a high-impact review. We have added a new chapter titled “Multidimensional Analysis of Key Bottlenecks in Nanomedicine Clinical Translation,” with each section supplemented by the latest evidence and practical insights(lines 1229-1305, page 31).

  1. b) Incorporation of Tables and Comparative Summaries

Although the text is comprehensive, the density of information may impede quick comprehension. The manuscript currently lacks comparative tables or schematic illustrations summarizing critical information such as nanomaterial classes, functionalization methods, targeting strategies, and therapeutic outcomes. Introducing well-structured tables and diagrams—such as mechanistic schematics for each therapy type and side-by-side comparisons of nanomaterial properties and efficacy data—would significantly improve readability, facilitate rapid reference, and broaden the review’s accessibility to a diverse scientific audience.

Response: We sincerely appreciate your valuable feedback regarding “excessive information density hindering rapid comprehension” and “adding comparative tables and schematic diagrams.” We fully agree that for a multidisciplinary review like “Progress in Nanomaterials for Tumor Therapy,” structured visualization tools are essential for distilling complex information and enhancing readability. They are also crucial for ensuring efficient information access across diverse scientific audiences, including material researchers, clinicians, and industry professionals. In the initial draft, we over-relied on textual descriptions to present key data (e.g., nanomaterial categories, functionalization methods, therapeutic outcomes). This led to scattered information and made it difficult for readers to quickly reference, a challenge that is particularly notable for cross-disciplinary audiences and aligns closely with the issues you highlighted. We have planned a supplementary scheme for core data visualization, which will outline the types, advantages and characteristics of common nanoparticles in tumor therapy through a summary table of nanomaterial-based combined tumor therapy and its features. (lines 669-676, page 17、lines 1091-1102, page 26).

  1. Consistency in Reference Title Capitalization

There are inconsistencies in the capitalization of reference titles (notably references 1, 6, 12, 17, 18, 23, 24, 32, 49, 54, 55, 69, 73, 74, 92, 95, 98, 144, 152, 159, 180, among others). Please standardize the formatting of all reference titles in accordance with the journal’s style guidelines to ensure professional and uniform presentation.

Response: We sincerely appreciate your detailed feedback regarding the inconsistent capitalization format in reference titles. To address this issue, we have made revisions strictly in accordance with the “Reference Format Requirements” outlined in the journal's Author Guidelines.

Round 2

Reviewer 3 Report

Comments and Suggestions for Authors

With the presented revisions, the authors comprehensively addressed all the raised points and considerably improved the manuscript. I am pleased to endorse this paper for acceptance in “Biomedicines”.

Reviewer 4 Report

Comments and Suggestions for Authors

This manuscript is recommended for acceptance.